# Neural DDEs with Learnable Delays for Partially Observed Dynamical Systems

## Abstract

Many successful methods to learn dynamical systems from data have recently been introduced. Such methods often rely on the availability of the system's full state. However, this underlying hypothesis is rather restrictive as it is typically not confirmed in practice, leaving us with partially observed systems. Utilizing the Mori-Zwanzig (MZ) formalism from statistical physics, we demonstrate that Constant Lag Neural Delay Differential Equations (ND-DEs) naturally serve as suitable models for partially observed states. In empirical evaluation, we show that such models outperform existing methods on both synthetic and experimental data. Code is available at https://anonymous.4open.science/r/DynamicalSysDDE-F86C/

## 1 Introduction

Learning system dynamics is essential in many domains such as biology (Roussel, 1996; Epstein, 1990), climate research (Ghil et al., 2008; Keane et al., 2019) or finance (Achdou et al., 2012). In a data-driven context, given a dataset of $L$ distinct trajectories of length $N + 1$, represented as $\{(t_0^l, x_0^l), \ldots, (t_N^l, x_N^l)\}_{l=1}^L$, which are observations of a unknown system:

$$\begin{aligned} \frac{\mathrm{d}x}{\mathrm{d}t} &= f(t, x(t)) \\ x(0) &= x_0 \end{aligned} \tag{1}$$

we wish to learn a model for the dynamics of $x(t)$ with $x : \mathbb{R} \to \mathbb{R}^n$ and $f : \mathbb{R} \times \mathbb{R}^n \to \mathbb{R}^n$. Using Equation 1's ODE dynamics model to fit the aforementioned dataset is natural and well suited, Neural Ordinary Differential Equations (NODEs), introduced in Chen et al. (2018), follows this exact formulation. NODEs gave rise to the class of continuous depth models and can be viewed as a continuous extension of Residual Networks (He et al., 2016). An immediate extension of NODEs, referred to as Augmented NODEs (Dupont et al., 2019), explores the existence of certain functions that NODEs are unable to represent. It tackles this expressivity limitation of NODEs by expanding the dimension of the solution space from $\mathbb{R}^n$ to $\mathbb{R}^{n+p}$ through the incorporation of an additional proxy variable $a(t)$. The additional introduced dimensions allow to learn more complex functions using simpler flows. The Augmented NODE is defined as follows:

$$\frac{d}{dt} \begin{bmatrix} x(t) \\ a(t) \end{bmatrix} = f\left(t, \begin{bmatrix} x(t) \\ a(t) \end{bmatrix}\right), \quad \begin{bmatrix} x(0) \\ a(0) \end{bmatrix} = \begin{bmatrix} x_0 \\ 0 \end{bmatrix}. \tag{2}$$

Unfortunately, many applications have to deal with partially observable states and non-Markovian dynamics making a NODE problem formulation ill-suited. An instance of this issue is in the metal cutting process, where the neglect of tools' vibration delay can lead to poor surface quality (Kalmár-Nagy et al., 2001). Similarly, ignoring non-Markovian phenomena such as gene transcription, translation times, and inter-compartmental transport within cells leads to unrealistic models of biological systems. Examples include gene expression, cell division, and circadian rhythms (Jensen et al., 2003; Tiana et al., 2007). Furthermore, in climate science, accurately predicting phenomena like the El Niño-Southern Oscillation (ENSO) necessitates the inclusion of feedback loops (Kondrashov et al., 2015). To address ODEs limited expressivity some sort of memory mechanism is required.

The distinctive feature of recurrent neural networks (RNNs) is their ability to incorporate a "memory" as a latent variable. This allows them to leverage past inputs, influencing both the current input and output (Jordan, 1986; Rumelhart et al., 1985; Hochreiter & Schmidhuber, 1997). RNNs are commonly employed for processing sequential or time series data, making them well-suited for dynamical systems. However, the discrete nature of RNNs collides with the appealing continuous formulation of the problem. Often compared to LSTMs, Reservoir Computing (RC), can be regarded as a noteworthy alternative to RNNs for its efficient training and strong performance in capturing long-term statistics when full state dynamics are accessible Vlachas et al. (2020). RCs use a fixed, randomly connected recurrent neural network (the reservoir) to capture the dynamics of input data while training only the output layer. Once mapped, only a simple readout layer is needed to extract the reservoir's state and train it to achieve the desired output Jaeger & Haas (2004); Maass et al. (2002)

An alternative hybrid technique named Latent ODE combines NODE and RNNs together (Rubanova et al., 2019). This variational-autoencoder model uses an ODE-RNN encoder and ODE decoder architecture to construct a continuous time model with a latent state defined at all times trained with a set of $N$ observations $x(t_i)$ at times $t_i$. The initial latent state $z_0$ is sampled via a parameterized Gaussian distribution. The final output (i.e., the entire trajectory) is generated by integrating the initial state $z_0$ over time with another parameterized vector-field $f_\theta$ and projecting it back onto the original space.

Another approach is to incorporate non-Markovianity into the formulation by including the historical past state in Equation 1. Such a choice promotes more transparency than RNNs rather "opaque" nature. This brings us into the domain of Neural Delay Differential Equations (NDDEs), which is another subset of models falling under the umbrella of continuous depth models, alongside NODEs. A generic delay differential equation (DDE) is described by :

$$
\begin{aligned}
&\frac{\mathrm{d}x}{\mathrm{d}t} = f(t, x(t), x(\alpha_1(t)), \ldots, x(\alpha_k(t))) \\
&\forall t,\ \alpha_i(t) = t - \tau_i(t, x(t)), \quad i \in 1, 2, \ldots, k \\
&x(t < 0) = \psi(t)
\end{aligned}
\tag{3}
$$

where, $\psi : \mathbb{R}^- \to \mathbb{R}^n$ is the history function, $\forall i, \tau_i : \mathbb{R} \times \mathbb{R}^n \to \mathbb{R}^+$ is a delay function, and $f : [0, T] \times \mathbb{R}^n \times \cdots \times \mathbb{R}^n \to \mathbb{R}^n$ can be a parameterized network. The history function $\psi$ serves as the initial condition for DDEs, analogous to $x_0$ in ODEs.

The modeling capabilities of NDDEs vary based on the chosen delay type. Inherently, NDDEs, with their delays, incorporate and leverage information from preceding time points, effectively converting the delay term into a dynamic memory mechanism. Initially proposed by Zhu et al. (2021) to learn NDDEs with a single constant delay, subsequent work by Zhu et al. (2023) and Schlaginhaufen et al. (2021) explored piece-wise constant delays and developed a stabilizing loss for NDDEs, respectively. Additionally, Oprea et al. (2023) focused on learning a single delay within a small network (less than 10 parameters).

In this paper, we extend these previous contributions by embedding NDDEs in the general framework of the MZ formalism. We explore the possibility of learning the values of the delays at the same time as neural flows for realistic network sizes and extend the state of the art to complex physical applications.

Our main contributions are the following :

- Provide a theoretically grounded formulation for partially observed systems with the Mori-Zwanzig formalism.

- Demonstrate that constant lag NDDEs can model partially observed dynamics with numerical examples and experimental data from fluid mechanics. Code is available at https://anonymous.4open.science/r/DynamicalSysDDE-F86C/.

- Provide an opensource package for constant lag DDEs compatible with neural networks, implemented in Pytorch. This implementation allows to learn jointly the delay and the DDE's dynamics. Code is available at https://anonymous.4open.science/r/torchdde-38F6/.

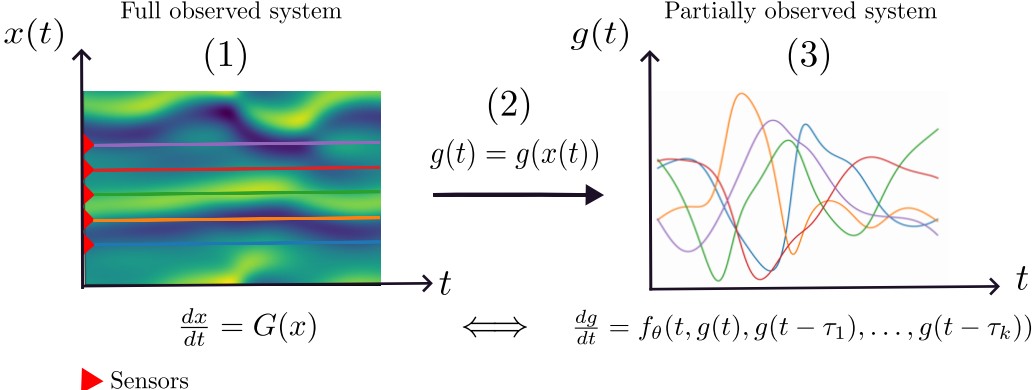

Figure 1: Measuring the fully observed state of a system (1) is often impossible due to its high dimensionality. Ultimately the user only has at its disposable sparse observations of the full state that can be seen as low dimensional observable function $g$ (2). The MZ equation DDE approximation (Proposition 2.2) can then used to model partially observed systems (3).

## 2 Modelling Partially Observed Dynamical Systems

In this section we first introduce the Mori-Zwanzig formalism, suitable for partially-observed dynamical systems, then discuss the limitations of the usual way to solve them in practice with Integro-Differential Equations, and propose our new approach based on Takens' theorem to tackle these issues.

### 2.1 The Mori-Zwanzig (MZ) formalism

The Mori-Zwanzig formalism, rooted in statistical mechanics, provides a method to construct accurate evolution equations for relevant quantities, such as macroscopic observables, within high-dimensional dynamical systems (Mori, 1965; Zwanzig, 1966; Zwanzig et al., 1972). This framework is instrumental in situations where the full state $x(t)$ is unavailable, and one can only access lower dimensional observations. Additionally, the MZ formalism is relevant for addressing dimension reduction problems (Zhu, 2019). We will operate within the first scenario, i.e. with partially-observed dynamical systems.

**Theorem 2.1.** *Mori-Zwanzig equation formalism* *Let us consider a nonlinear system evolving on a smooth manifold $\mathcal{S} \subset \mathbb{R}^n$:*

$$\frac{\mathrm{d}x}{\mathrm{d}t} = G(x), \quad x(0) = x_0, \tag{4}$$

*where the full state $x \in \mathcal{S}$ can be accessed only through the lens of an arbitrary number of scalar-valued observables $g_i : \mathcal{S} \to \mathbb{R}$. Then, the dynamics of the vector of observables $g = [g_1, \ldots, g_m]$ follow the Integro-Differential Equation (IDE) :*

$$\frac{dg}{dt} = M(g(t)) - \int_0^t K(g(t-s), s)ds + F(t). \tag{5}$$

*where $M$ is a Markovian term, i.e. a function involving current observables $g(t)$, and where the function $K$, which depends on past observables, is integrated over the whole time since the initial condition, while the noise term $F$ accounts for the dynamics of the observables in the unobservable space.*

This equation, derived within the framework of the Mori-Zwanzig theory (see Appendix A), is sometimes referred to as the Generalized Langevin Equation (GLE). It provides a rigorous governing equation for the observables $g$. This elegant formulation of partially observed dynamics yields an exact evolution equation that takes the form of an Integro-Differential Equation. Therefore, learning the dynamics of partially observed systems boils down to estimating each term of the differential equation above. Appendix B demonstrates some cases where the noise term $F$ can be null, a scenario we set ourselves in.

## 2.2 Approximations of Integro-Differential Equations (IDE)

The statement in Theorem 2.1 outlines the structure of the vector field dynamics. However, solving this dynamical equation can prove challenging due to the complexity implied by the integral term. We now discuss several approaches from the literature to estimate this integral.

A simplistic approximation consists in just disregarding the integral term, focusing solely on the impact of the observable $g$ at the current time step $t$. This corresponds to NODE Chen et al. (2018), where the dynamics are approximated as:

$$\frac{\mathrm{d}g}{\mathrm{d}t} \approx f_\theta(t, g(t)).$$

where $f_\theta$ is a neural network with parameters $\theta$.

Instead of neglecting it, an approach to approximate the integral consists in studying particular asymptotic regimes. Among the many different models proposed in the literature (Stinis, 2003; Chorin & Stinis, 2005; Stinis, 2006), one of the the most popular consists in approximating the integral under assumptions of very short memory or very long memory regimes.

For example, the *t-model*, also commonly called *slowly decaying memory approximation* (Chorin et al., 2002) leads to Markovian equations with time-dependent coefficients, , which can subsequently be modeled using a NODE (see Appendix C for full derivation). However these remain asymptotic approximations, and, more problematically, they are not extendable to intermediate-range memory in general.

Another approach consists in performing Monte Carlo integration (Robert et al., 1999). This work has been extended in a neural network-based formulation (Neural IDE) Zappala et al. (2022), where the memory integrand is decomposed as a product of type $K(t, s)F(g(s))$. However the number of function evaluations required to accurately integrate Equation 5 scales with the increasing value of $t$, making the process computationally intensive. In practice, experiments indicate that Neural IDE is at *least 150 times slower* compared to other models introduced subsequently (cf Appendix E). In a similar spirit, under assumptions of short memory, one can restrict the integral to a short past and discretize it in time, leading to an equation of the form (Gallage, 2017):

$$\frac{dg}{dt} \approx M(g(t)) - \frac{1}{k} \sum_{i=1}^{k} K(g(t - \tau_i), \tau_i) \tag{6}$$

using $k$ delays $\tau_i$ uniformly spaced instead of sampling them by Monte Carlo. Such approximations can be improved using high-order discretization schemes, yet, as for Neural IDE, they require an unaffordable number of delays $\tau_i$ if the integrand varies quickly or if the interval is too large.

## 2.3 Exact representation with Neural DDE

While Equation (6) only provides an approximation of the true dynamics and requires many delays, we show that using a more complex function of a small number of delays it is actually possible to represent the dynamics *exactly*. For this, let us consider diffeomorphic dynamical systems, that is, ODEs whose flow is invertible (in the full state space) and smooth in both time directions.

**Proposition 2.2 (Exact representation with delays).** *For any diffeomorphic dynamical system, and differentiable observables $g$, using the same notations as for Theorem 2.1, there exists almost surely a function $M$ of the current observables, a finite number $k$ of delays $\tau_1, ..., \tau_k > 0$ and a function $f$ such that the observables* exactly *follow the dynamics:*

$$\frac{dg}{dt} = M(g(t)) + f\big(t, g(t), g(t - \tau_1), g(t - \tau_2), ..., g(t - \tau_k)\big). \tag{7}$$

The proof, deferred to Appendix I, is based on the application of Takens' theorem (hence the diffeomorphism requirement), which also provides a bound on the required number of delays: twice the intrinsic dimension of the manifold $\mathcal{S}$ in which the full state $x$ lives, plus one. Note that this evolution equation is exact:

approximations may arise from the optimization or the expressivity of the neural networks estimating $M$ and $f$, but not from the number of delays provided they reach Takens' bound. This is in contrast with the discretization of the IDE integral as in Equation (6), which becomes asymptotically precise only when the number of delays becomes large compared to the complexity of the integrand. Note that integral discretizations as in Equation (6) are doable with a single linear layer of a neural network taking past observables $g(t - \tau_i)$ as input, and that Proposition 2.2 actually states that by stacking more layers one can reach exact representation of the dynamics.

Additionally, this framework is also motivated in practice with climate models from Ghil et al. (2008); Falkena et al. (2021) that utilize the MZ formalism to derive DDE structures.

Experiments in Section 4 will illustrate that NDDEs where *both the delays and their dynamics are learned jointly* can effectively capture the dynamics of partially observed systems. Before this, we detail how to perform such a training in the next section. In Figure 1, the general scenario is highlighted, wherein users have access solely to the system's observables and we wish to learn their dynamics by using the *MZ equation* approximation (Eq. 7).

## 3 Neural Delay Differential Equations with Learnable Delays

A constant lag NDDE is part of the larger family of continuous depth models that emerged with NODE Chen et al. (2018), it is defined by:

$$
\begin{aligned}
\frac{\mathrm{d}x}{\mathrm{d}t} &= f_\theta(t, x(t), x(t - \tau_1), \ldots, x(t - \tau_k)) \\
x(t < 0) &= \psi(t)
\end{aligned}
\tag{8}
$$

where $\psi : \mathbb{R} \to \mathbb{R}^n$ be the history function, $\forall i, \tau_i \in \mathbb{R}^+$ be a delay constant and $f_\theta : [0, T] \times \mathbb{R}^n \times \cdots \times \mathbb{R}^n \to \mathbb{R}^n$ be neural network.

There are two possible ways of training continuous-depth models: discretize-then-optimize or optimize-then-discretize (Kidger, 2022). In the former, the library's inherent auto-differentiation capabilities are leveraged. In the latter, the adjoint dynamics are employed to compute the gradient's loss. Theorem 3.1 provides the adjoint method for constant lag NDDEs.

**Theorem 3.1.** *Let us consider the continuous-depth DDE model below where $\tau$ can appear in the parameters vector $\theta$ and the notation $x(t) = x_t$ for conciseness :*

$$
\begin{aligned}
x'_t &= f_\theta(t, x_t, x_{t-\tau}), \quad \tau \in \mathbb{R}^+ \\
x_{t<0} &= \psi_t
\end{aligned}
\tag{9}
$$

*and the following loss function :*

$$
L(x_t) = \int_0^T l(x_t) \, dt
$$

*The gradient's loss w.r.t. the parameters is given by:*

$$
\frac{\mathrm{d}L}{\mathrm{d}\theta} = -\int_0^T \lambda_t \left( \frac{\partial f(x_t, x_{t-\tau}; \theta)}{\partial \theta} + \frac{\partial f(x_t, x_{t-\tau}; \theta)}{\partial x_{t-\tau}} x'_{t-\tau} \right) dt + \int_{-\tau}^0 \lambda_{t+\tau} \frac{\partial f(x_{t+\tau}, x_t; \theta)}{\partial x_t} \frac{\partial \psi_t}{\partial \theta} dt.
\tag{10}
$$

*where the adjoint dynamics $\lambda_t$ are given by another DDE:*

$$
\begin{aligned}
\dot{\lambda}_t &= \frac{\partial l(x_t)}{\partial x_t} - \lambda_t \frac{\partial f(x_t, x_{t-\tau}; \theta)}{\partial x_t} - \lambda_{t+\tau} \frac{\partial f(x_{t+\tau}, x_t; \theta)}{\partial x_t}, \\
\lambda_{t \geq T} &= 0.
\end{aligned}
\tag{11}
$$

Proof for Theorem 3.1 and a generalization for multiple constant delays are in Appendix G. The second integral term in equation 10 is more often than not null since the history function $\psi_t$ is user defined and not learnt. Algorithm 1 that uses Theorem 3.1's results outlines the training procedure for a Neural DDE with one learnable constant delay (Appendix H provides a version for multiple delays). Without loss of generality, we set ourselves in the case where the dataset $\mathcal{D}$ is comprised of one trajectory:

---

**Algorithm 1** Training a Neural DDE with one learnable delay with the adjoint method.

---

**Require:** Dataset of one trajectory $\mathcal{D} = \{(t_0, x_0), \ldots, (t_N, x_N)\}$.
**Require:** Initialized model $f_\theta$.
**Require:** Initialized positive delays $\tau$ that can appear in the parameters vector $\theta$.
1: **for** $i \leftarrow 1, \ldots, N_{epochs}$ **do**
2:   Create history function interpolation $\psi$ with data from $\mathcal{D}$ such that $t < \tau$.
3:   Solve DDE dynamics (Eq. 9):
4:   $\begin{cases} x'(t) = f_\theta(t, x(t), x(t - \tau)) \\ x(t < \tau_{max}) = \psi(t) \end{cases}$ .
5:   Compute loss $L(x(t_N)) = \int_{\tau_{max}}^{t_N} l(x(s)) \, ds$
6:   Solve Adjoint dynamics (Eq. 11):
7:   $\begin{cases} \lambda'(t) = \frac{\partial l(x(t))}{\partial x(t)} - \lambda(t) \frac{\partial f_\theta(x(t), x(t-\tau))}{\partial x(t)} - \lambda(t + \tau) \frac{\partial f_\theta(x(t+\tau), x(t))}{\partial x(t)} \\ \lambda(t \geq t_N) = 0. \end{cases}$ .
8:   Compute $\frac{dL}{d\theta}$ (Eq. 10)
9:   Update $\theta$
10: **end for**

---

Learning the delays $\tau_i$ within NDDEs is crucial for accurately modeling partially observed dynamics. Indeed, certain delays may not provide relevant enough information (if too small, entries of the delay vector data are too similar; if too large, the entries tend to be completely uncorrelated and cannot be numerically linked to a consistent dynamical system), therefore delays need to be adapted during training. This is illustrated in the second **Cavity** experiment in Section 4.2, where the training of NDDEs with fixed delays remains stuck, while concurrently learning the delays converges to a satisfactory solution. Appendix J briefly illustrates how suitable delays (and the information contained within these delayed observables) can impact the model's learning process.

Substantial efforts have been expanded to design a user-friendly API, developing a numerically robust DDE solver, and implementing the adjoint method (referenced as Theorem 3.1) in the `torchdde` package. These advancements ensure a seamless integration of DDEs for future users, enhancing reproducibility. Further details about `torchdde` are available in the supplementary materials and benchmarks are found in Appendix E.3.

## 4   Experiments

Numerous experiments have been carried out, categorically addressing two aspects: firstly, validating the existing adjoint approach and assessing the benefits of incorporating learnable constant delays; secondly, examining how neural DDEs with learnable delays is essential to effectively approximate partially observed systems, demonstrated on both synthetic data and experimental data. All datasets are divided into training, validation, and test sets with proportions of 70%, 10%, and 20%, respectively. Additional experiments are also provided in Appendix F.

In summary, the experiments involving the Brusselator and KS System presented below demonstrate the learning of multiple delays. NDDE is the only model that accurately captures the statistics of the KS System. In the Cavity experiment presented below, learning delays is crucial, as fixed delays fail to capture the system's dynamics correctly.

### 4.1 Dynamical systems

**Toy Dataset** We demonstrate that the current approach with the adjoint method can learn jointly the delay and the dynamics of a system used to model population dynamics in biology Arino et al. (2009); Banks et al. (2017). Such a described system is formulated through the following DDE :

$$\frac{\mathrm{d}x}{\mathrm{d}t} = x(t)\left(1 - x(t - \tau)\right),$$
$$x(t < 0) = \psi(t) \tag{12}$$

where we integrate from $t \in [0, 10]$, $\tau = 1$, $\psi(t) = x_0$, $x_0$ is sampled from the uniform distribution $\mathcal{U}(2.0, 3.0)$ adn 256 trajectories were generated.

The following experiments showcase how NDDEs can effectively model partially observed systems with the systems past state values rather than with opaque latent variables.

**Brusselator** The Belousov-Zhabotinsky kinetic equation Belousov (1959); Zhabotinskii (1964) can be modelled by the Brusselator system :

$$\begin{cases} \frac{\mathrm{d}\phi_1}{\mathrm{d}t} = A - B\phi_1 - \phi_1 + \phi_1^2\phi_2 \\ \frac{\mathrm{d}\phi_2}{\mathrm{d}t} = B\phi_1 - \phi_1^2\phi_2. \end{cases} \tag{13}$$

where we integrate in the time domain $t \in [0, 25]$, the initial condition $\phi_1$ is sampled from the uniform distribution $\mathcal{U}(0, 2.0)$, $\phi_2 = 0.0$ and 1024 trajectories were generated . We set ourselves in the partially observable case where we only have $\phi_1$'s dynamics and wish to reconstruct its dynamics.

**Kuramoto–Sivashinsky (KS) System** We set ourselves in another experiment with the chaotic Kuramoto–Sivashinsky System whose 1D dynamics $u(x, t)$ is :

$$\frac{\partial u}{\partial t} + \frac{\partial^2 u}{\partial x^2} + \frac{\partial^4 u}{\partial x^4} + \frac{1}{2}\frac{\partial u^2}{\partial x} = 0$$

The system is integrated over the time domain $t \in [0, 30]$, its spatial domain $D_x = [0, 22]$ is discretized into 128 points and 2048 trajectory samples were generated. To put ourselves in the partially observed setting we choose to observe $k$ features uniformly spread across the spatial domain (here $k = 5$).

**Incompressible open cavity flow** We consider here as experimental demonstrator the modelling based on time-series derived from wind tunnel experiments of an open cavity flow represented in Figure 2; the facility is described in Tuerke et al. (2020), where the data we provide in open access with this work are discussed. Open cavity flow attracted numerous research efforts in the last decades for the interesting dynamics at work: the flow is characterized by an impinging shear layer activating a centrifugal instability in a cavity; this interplay, reminiscent of the feedback acoustic mechanisms described in Rossiter (1964), leads to a self-sustained oscillation. A broad range of dynamics is observed, ranging from limit cycles, to toroidal and chaotic dynamics. The data obtained is for a Reynolds number $Re = 9190$. More details on the experimental setup is given in Tuerke et al. (2020).

| | LSTM | NODE | ANODE | Latent ODE | NDDE |
|---|---|---|---|---|---|
| Brusselator | **0.0051 ± 0.0031** | 0.77 ± 0.00080 | **0.0050 ± 0.0050** | 0.014 ± 0.0076 | 0.016 ± 0.0076 |
| KS | 0.77 ± 0.041 | 0.71 ± 0.10 | 0.55 ± 0.027 | 0.43 ± 0.07 | **0.28 ± 0.024** |
| Cavity | 0.75 ± 0.46 | 0.96 ± 0.0011 | 0.65 ± 0.0090 | 0.25 ± 0.14 | **0.13 ± 0.012** |

Table 2: Model performance (MSE) over the test set on each experiments averaged over 5 runs

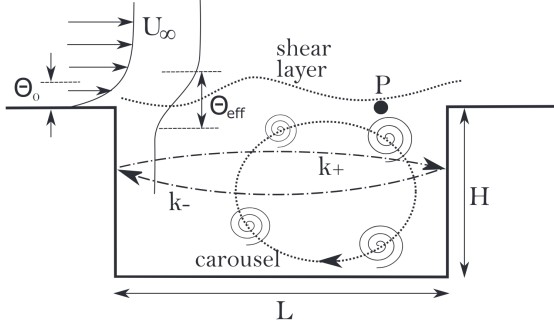

Figure 2: Sketch of open cavity flow taken from Tuerke et al. (2020). A data acquiring sensor is placed in P. The cavity has a length of $L$ and depth of $H$. The incoming laminar boundary layer is characterized by the freestream velocity $U_\infty$ and the momentum thickness $\Theta_0$.

| | KS | Cavity | Brusselator |
|---|---|---|---|
| NDDE | 5 | 1 | 2 |

Table 1: Number of delays used in NDDE for each experiment

## 4.2 Results

In this section, we assess the performance of the models with their ability to predict future states of a given partially observed system along with the toy dataset experiment. In this study, LSTM, NODE, ANODE, Latent ODE and NDDE were selected for comparison, and Table 2 displays the test MSE loss over each experiment. Appendix K goes in more details about each model's architecture and the training and testing procedure. Every model incorporates a form of 'memory' into its architecture, with the exception of NODE. While LSTM and Latent ODE utilizes hidden units and ANODE employs its augmented state $a(t)$, NDDE leverages past states such as $x(t-\tau)$. Finally, Table 1 outlines the number of delays employed in NDDE for each experiment. We provide in Appendix D a general discussion on the delays learnt and their evolution during the training process of each experiment. In all subsequent figures, the y-axis $y(t)$ represents our observables (introduced for each system in Section 4.1), defined as $y(t) = g(x(t))$.

**Toy dataset** Figure 3 & 4 respectively depict the model's robust convergence to accurate dynamics and the delay evolution during training over many seeds, showcasing a consequence of Takens' theorem (Takens, 1981), that is, by using a delay-coordinate map, one can construct a diffeomorphic shadow manifold $M'$ from univariate observations of the original system in the generic sense. In our case such a lag variable is $(x(t), x(t-\tau))$ with $\tau \in \mathbb{R}^+$. The result of Figure 4 may seem surprising as the underlying DDE has a unique delay $\tau = 1$. However, we are not approximating the exact dynamics from the DDE itself but rather from the shadow manifold $M'$. In classical approaches (see Tan et al. (2023a;b)), the selected delay with Takens' theorem for SSR corresponds to the time series' minimum delayed mutual information which measures the general dependence of two variables Fraser & Swinney (1986).

---

[0]Figure taken from Tuerke et al. (2020)

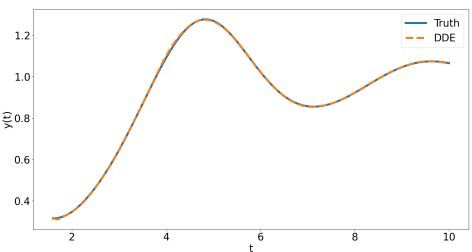

Figure 3: Toy dataset random test sample

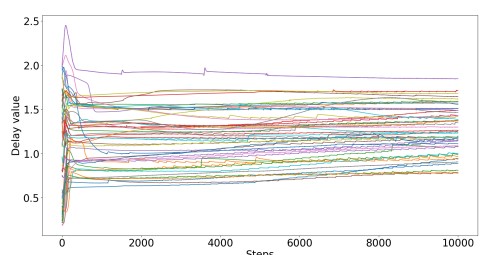

Figure 4: Toy dataset delay evolution during training

**Brusselator** In the case of this highly stiff and periodic dataset, all models demonstrate satisfactory performance except for NODE. NODE predicts the trajectory's mean thus highlighting the importance of incorporating memory terms. Remarkably, both LSTM and ANODE perform equally well, with NDDE and Latent ODE slightly trailing by a narrow margin as shown in Figure 5. In addition to evaluating the MSE loss performance, Figure 6 demonstrates the stability of each trained model on the Brusselator system over an extended period. After training within a specific time interval, we lengthened the integration period to five times the original training duration to assess the models' performance. It is observed that NDDE, along with LSTM, NODE and Latent ODE, are the only models that remain stable throughout this duration, with NDDE exhibiting the best performance over the extended horizon.

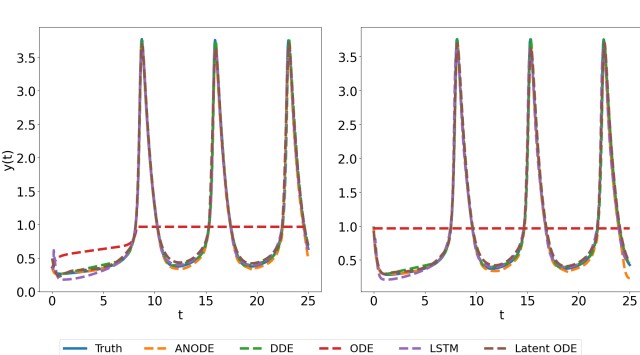

Figure 5: Brusselator random test sample

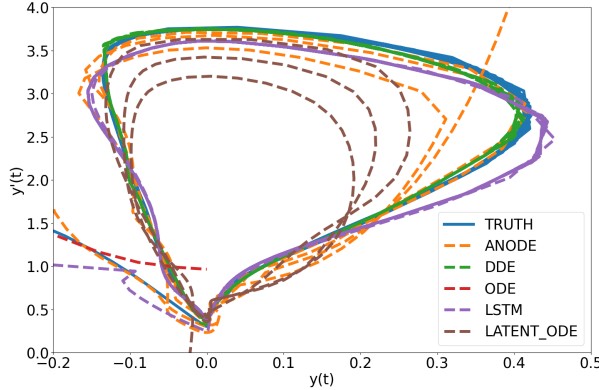

Figure 6: The long-term behavior of each trained model for the Brusselator system

**KS System** This experiment deals with a chaotic setting of the partially observed system. By observing periodically $k$ features, the hopes of estimating the high order spatial partial derivatives of $u(x, t)$ is in vain making such formulation of the problem even more challenging. Figure 7 showcases random test samples from two different training runs, highlighting how NDDEs outperform other models struggling with the dynamics of the selected features. Furthermore, in a chaotic setting, the statistics of the dynamics prove more informative than the trajectory itself. Figure 8 demonstrates that the DDE formulation is distinctive in its adherence to the density distribution of the KS System's test set. This outcome highlights that utilizing past states $x(t - \tau)$ as memory buffers, as opposed to opaque latent variables, can lead to superior results. Furthermore, considering the chaotic nature of the system, we calculated an important metric: the maximum Lyapunov exponent (MLE) of the generated trajectories from our trained models. Table 3 displays the MLE estimates for each model, showing that the Neural DDE with learnable delays closely aligns with the ground truth compared to other models.

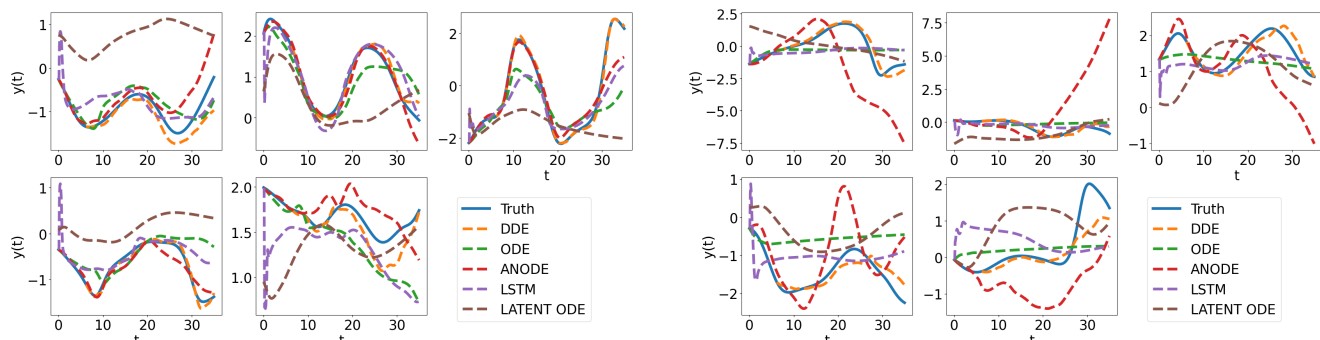

Figure 7: KS random test sample

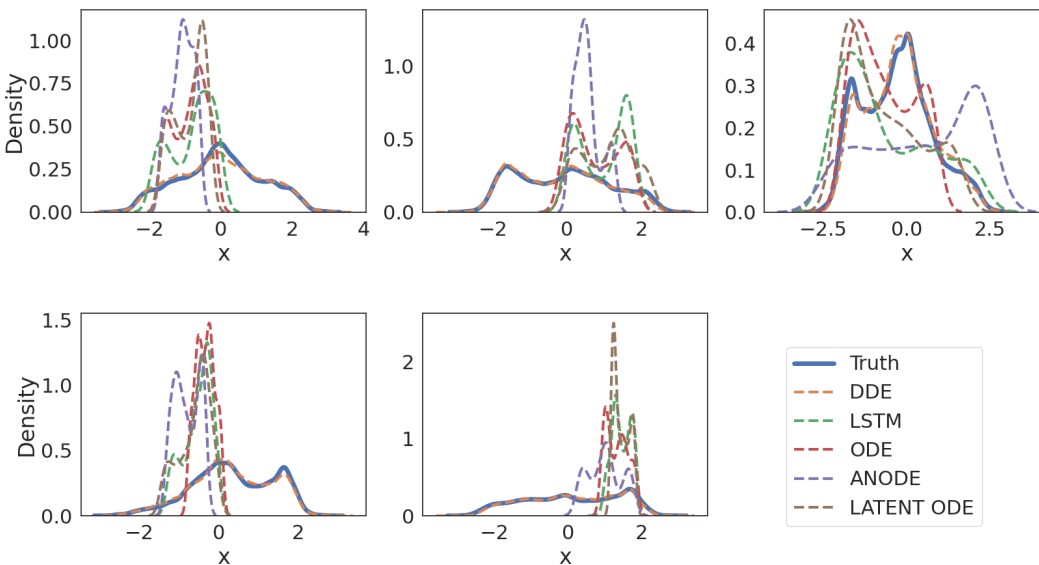

Figure 8: Example of KS testset density plots

|  | Ground Truth | NDDE | NODE | ANODE | Latent ODE |
|---|---|---|---|---|---|
| $\lambda_{max}$ | 0.1291 | **0.1279** | 0.0968 | 0.1198 | 0.0354 |

Table 3: Estimation of the maximum Lyapunov exponent $\lambda_{max}$ for the KS system based on the generated trajectories from the test set for each model.

**Cavity** Once more, the NDDE formulation distinguishes itself from other models as seen in Figure 9, and this can be attributed to various factors. The experimental setup encourages a delayed formulation of the problem, with the vortex-induced flow originating from the cavity, coupled with partial observability issues. Latent ODE yields acceptable results compared to NODE that generates the system's average trajectory, while LSTM and ANODE capture vague oscillations, albeit occasionally in conflicting phases. Finally, these experiments demonstrate that NDDE can effectively model trajectories even in the presence of noise in the data.

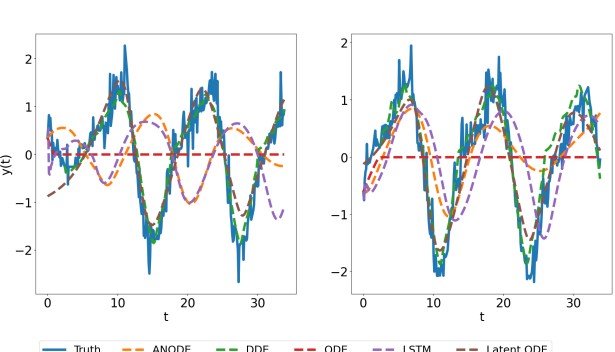

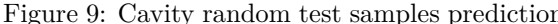

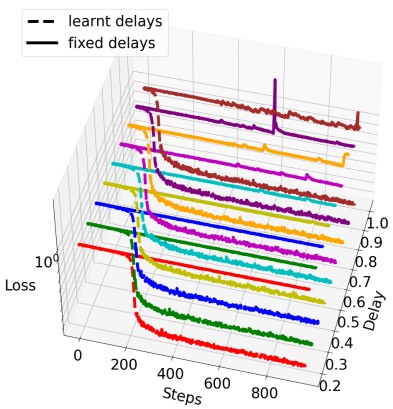

Figure 9: Cavity random test samples prediction

Figure 10: NDDE's with constant and learnable delays MSE train loss for different delay initialization values ranging from 0.2 to 1.0.

Lastly, we compare NDDEs with learnable and fixed delays on our Cavity dataset. Figure 10 illustrates how the value of initial delays affects system learning. Delays are initialized from 0.2 to 1.0 in increments of 0.1, with the models starting with the same weights. Dotted lines represent the setup with learnt delays, while solid lines indicate fixed delays. The results show that learnt delays consistently outperform fixed delays. Additionally, the transition from a learnt delays loss of $10^0$ to $10^{-1}$ can occur more easily depending on the initial value. For instance, a delay initialized at 1.0 is seen to converge faster than those initialized at 0.2.

## 5 Conclusion

In this study, we showcased the capability of constant lag neural delay differential equations (NDDEs) to effectively represent partially observed systems. The theoretical support for this assertion comes from the Mori-Zwanzig formalism and with its simplification that introduces DDE dynamics. We applied NDDEs to synthetic, chaotic, and real-world noisy data, and conducted comparisons with other continuous-depth and memory based models. The performed experiments revealed two key insights: firstly, the essential role of memory in accurately capturing dynamics; secondly, it was demonstrated that LSTMs' and Latent ODEs' hidden latent states or ANODEs' latent variables are not the exclusive means or sometimes come short to achieve optimal performance, emphasizing the efficacy of delayed terms as an efficient dynamic memory mechanism.

NDDEs come with inherent limitations, such as the linear scaling of its adjoint method with the number of delays (refer to Appendix G for the case of multiple constant delays). Another question is how to determine the optimal number of delays to consider; this said, overestimating the number of delays does not hurt the final performance. Promising directions for future research involve exploring an equivalent version of ANODEs with NDDEs to assess whether simpler flows can be learned. Additionally, there is a research opportunity to investigate regularization terms that could enhance the NDDE training process. Specifically, we are contemplating the inclusion of a penalty term resembling of a delayed mutual information, inspired by the work of (Fraser & Swinney, 1986).

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

## A  Derivation of Theorem 2.1

The dynamics of a physical model can be written as the evolution equation of the form

$$\frac{\partial}{\partial t} e^{t\mathcal{G}} u(0) = \mathcal{G} u(0)$$

where the $e^{t\mathcal{G}}$ is the evolution operator and $\mathcal{G}$ is the corresponding infinitesimal generator and $u$ an observable function Zhu (2019). The goal is to find an appropriate projector $\mathcal{P}$ and $\mathcal{Q} = \mathcal{I} - \mathcal{P}$ that splits the dynamics of the original high-dimensional system into resolved variables, unresolved variables and the interaction between these two.

If we consider a nonlinear system evolving on a smooth manifold $\mathcal{S} \subset \mathbb{R}^n$.

$$\frac{dx}{dt} = F(x), \quad x(0) = x_0$$

The system can be seen through the lens of an arbitrary number of scalar-valued observables $\forall i, g_i : \mathcal{S} \to \mathbb{C}$. The dynamics of any scalar-valued observable $g_i(x)$ (quantity of interest) can be expressed with the Koopman operator $\mathcal{K}(t, s)$ Koopman (1931),

$$g_i(x(t)) = [\mathcal{K}(t, s) g_i](x(s)) \tag{14}$$

$$\mathcal{K}(t, s) = e^{(t-s)\mathcal{L}}, \quad \mathcal{L} g_i(x) = F(x) \cdot \nabla g_i(x) \tag{15}$$

with $\mathcal{L}$ the Liouville operator. Often rather than not, instead of computing the dynamics of all observables, it is better to compute the evolution of a subset of quantities of interest. This subspace can be modelled with a bounded linear operator $\mathcal{P}$ (projector) and its orthogonal projector $\mathcal{Q} = \mathcal{I} - \mathcal{P}$. Since we are only considering a subset of observable we seek to get the dynamics of $\mathcal{P}\mathcal{K}(t, s)$. With the definition of the Koopman operator and the Dyson identity

$$e^{t\mathcal{L}} = e^{t\mathcal{Q}\mathcal{L}} + \int_0^t e^{s\mathcal{L}} \mathcal{P}\mathcal{L} e^{(t-s)\mathcal{Q}\mathcal{L}} ds \tag{16}$$

we obtain the Mori-Zwanzig operator equation

$$\frac{d}{dt} e^{t\mathcal{L}} = e^{t\mathcal{L}} \mathcal{P}\mathcal{L} + e^{t\mathcal{Q}\mathcal{L}} \mathcal{Q}\mathcal{L} + \int_0^t e^{s\mathcal{L}} \mathcal{P}\mathcal{L} e^{(t-s)\mathcal{Q}\mathcal{L}} \mathcal{Q}\mathcal{L} ds \tag{17}$$

The three terms at the right hand side are respectively the streaming (or Markovian) term, the fluctuation (or noise) term and the memory term.

Then, one can apply Eq. equation 17 to an arbitrary observable function $g_i$ and evaluate it at $x$ to get the dynamics of the projected (or not) resolved variables of the system.

One can rewrite also the MZ's functional form equation (Tian et al., 2021) by applying equation 17 to all observables $g_i$ at their initial condition $g_i(t = 0) = g_{i0}$ and concatenating all observable $g_i$ together into $g = [g_1, \ldots, g_m]$.

$$\frac{dg(t)}{dt} = M(g(t)) + F(t) - \int_0^t K(g(t-s), s) ds \tag{18}$$

where

$$M(g(t)) = e^{t\mathcal{L}}\mathcal{P}\mathcal{L}g_0 \tag{19}$$

$$F(t) = e^{t\mathcal{Q}\mathcal{L}}\mathcal{Q}\mathcal{L}g_0 \tag{20}$$

$$K(g(t-s),s) = -e^{(t-s)\mathcal{L}}\mathcal{P}\mathcal{L}e^{s\mathcal{Q}\mathcal{L}}\mathcal{Q}\mathcal{L}g_0 \tag{21}$$

## B  Cancelling out the noise term $F(x,t)$

One possibility is by applying the projector $\mathcal{P}$ we get rid of the fluctuation/noise term.

$$\frac{\partial}{\partial t}\mathcal{P}e^{t\mathcal{L}} = \mathcal{P}e^{t\mathcal{L}}\mathcal{P}\mathcal{L} + \int_0^t \mathcal{P}e^{s\mathcal{L}}\mathcal{P}\mathcal{L}e^{(t-s)\mathcal{Q}\mathcal{L}}\mathcal{Q}\mathcal{L}ds \tag{22}$$

since the $e^{t\mathcal{Q}}\mathcal{Q}\mathcal{L}$ and $\mathcal{P}$ live in orthogonal subspaces. Instead of learning your observable $g$ you consider $\mathcal{P}g$.

The other option depends on information that is unavailable as it is orthogonal to the observed subspace. However, this term vanishes if the history of the observed subspace is known, and the orthogonal dynamics are dissipative Menier et al. (2023).

## C  t-model derivation

In the case of the slowly decaying memory (i.e., *t-model*), we have the approximation :

$$e^{t\mathcal{Q}\mathcal{L}} \approx e^{t\mathcal{L}}. \tag{23}$$

By rewriting the memory term of the Mori-Zwanzig equation (Eq. 17)

$$\int_0^t e^{(t-s)\mathcal{L}}\mathcal{P}\mathcal{L}e^{s\mathcal{Q}\mathcal{L}}\mathcal{Q}\mathcal{L}ds = \int_0^t \mathcal{L}e^{(t-s)\mathcal{L}}e^{s\mathcal{Q}\mathcal{L}}\mathcal{Q}\mathcal{L}ds - \int_0^t e^{(t-s)\mathcal{L}}e^{s\mathcal{Q}\mathcal{L}}\mathcal{Q}\mathcal{L}\mathcal{Q}\mathcal{L}ds.$$

where we used the commutation of $\mathcal{L}$ and $\mathcal{Q}\mathcal{L}$ with $e^{t\mathcal{L}}$ and $e^{s\mathcal{Q}\mathcal{L}}$, respectively. By using Equation 23, which eliminates the $s$ dependence of both integrands we get :

$$\int_0^t e^{(t-s)\mathcal{L}}\mathcal{P}\mathcal{L}e^{s\mathcal{Q}\mathcal{L}}\mathcal{Q}\mathcal{L}ds \approx te^{t\mathcal{L}}\mathcal{P}\mathcal{L}\mathcal{Q}\mathcal{L}.$$

Only time dependence remains of the memory integral. We refer to Zhu (2019) for a more detailed discussion on the *t-model*.

## D  Learning the delays

As a reminder, Table 1 provides the number of delays used in each experiment.

For each experiment, we present randomly selected models and display the evolution of delays during training in Figure 11. Empirically, across all our experiments, we observe that the delays converge to specific values. Notably, if the system is periodic, these values do not match the system's period as observed in the Brusselator experiment. This makes sense, as incorporating such a delay would not provide any additional information to the NDDE model. For the KS system experiments, we see that the delays that are initialized close to each tend to spread out during the training phase in order to maximize the system's information diversity.

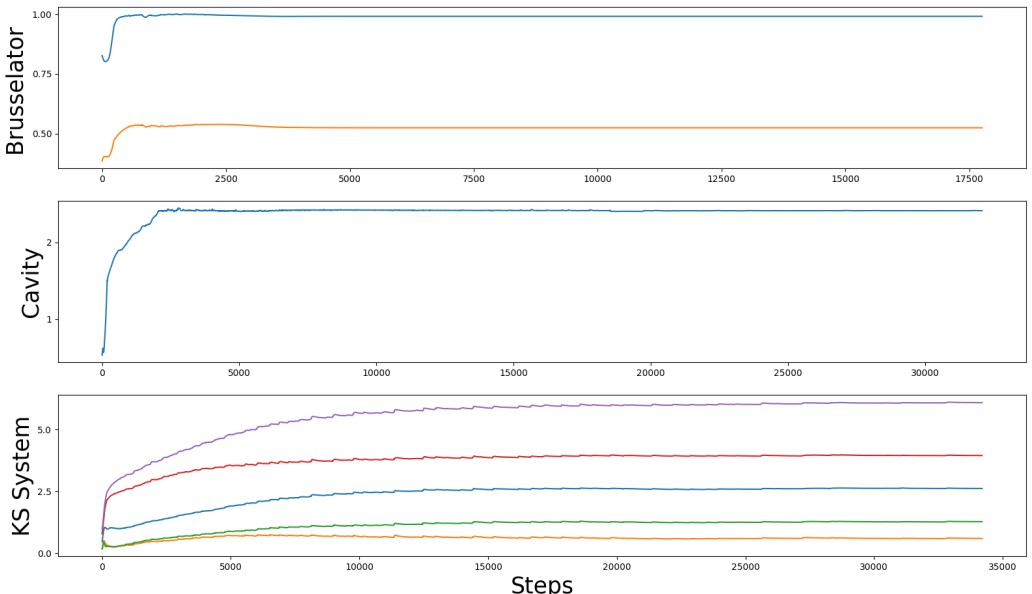

Figure 11: Delay's evolution during training for each experiment mentioned on each subplot's y-axis

# E    Neural IDE and Neural DDE Benchmark

Firstly, let us compare both Neural IDE and Neural DDE analytically where any function $f_\theta$ denotes a parameterized network:

$$\frac{dg}{dt} = M_\theta(g(t)) - \int_0^t K_\theta(g(t-s), s)ds \tag{24}$$

$$\frac{dg}{dt} = f_{\theta_1}(g(t)) + f_{\theta_2}(t, g(t), g(t-\tau_1), \dots, g(t-\tau_n)) \tag{25}$$

The second term on the right hand side of Equation 24 is much more computationally involved than that of the second term on the right hand side of Equation 25 . Indeed, Equation 25 only needs 2 function evaluations to evaluate it's right hand side(RHS). On the other hand, the number of function evaluation required to integrate Equation 24 will scale as $t$ grows in order to get a correct evaluation of the integral term. For a more theoretical examination of computational complexity, please refer to Appendix A.6 of Monsel et al. (2024).

## E.1    Computation Time

Figure 12 compares the computation time of a forward pass between a Neural IDE and Neural DDE of the same size (roughly 500 parameters). We use the following setup :

- Use a RK4 solver with a timestep of $dt = 0.1$.

- Use a batch size of 128.

- Neural DDE uses 5 delays.

- Different state $g(t)$ dimensions are tested : $[5, 10, 50, 100]$ .

- The upper integration bound is varied from $t = 0.2$ to $t = 2.0$s.

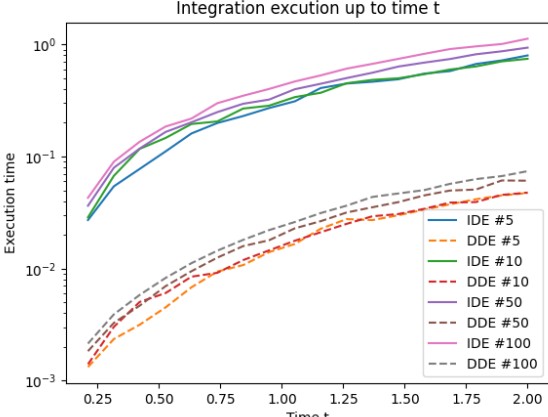

Figure 12: Time duration of forward pass averaged over 5 runs

This benchmark clearly shows how expensive Neural IDE is. NDDEs integration is at least an order of magnitude faster. (Please note that in Figure 12, the notation "#i" refers to the number of features that our state $g(x, t)$ has).

## E.2 Memory Consumption

We also performed memory profiling between the two methods and compared their memory needs for one forward pass. We use the following setup :

- Use a RK4 solver with a timestep of $dt = 0.1$.

- Use a batch size of 128.

- Neural DDE uses 5 delays.

- Different state $g(t)$ dimension is 100 .

- The upper integration bound is varied from $t = 0.2$ to $t = 2.0$s.

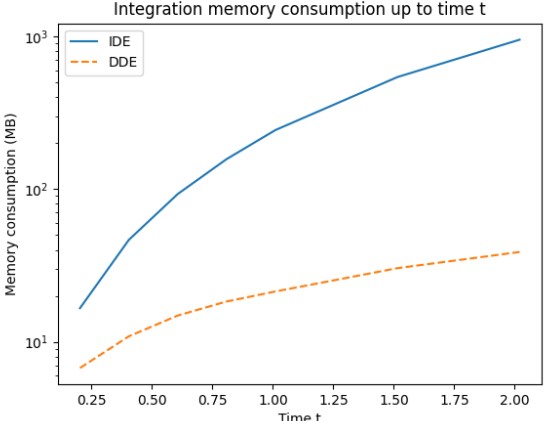

Figure 13: Memory consumption of forward pass averaged over 5 runs

For extremely small tasks, the memory requirements of IDEs are excessive, as depicted in Figure 13. Despite this, we attempted to implement the Neural IDE method in our experiments. Unfortunately, during training, we encountered memory issues, even with a simple problem like the Brusselator. The reason behind this is that the integral component needs to be recalculated at each integration step, causing scalability problems when the integration duration is extensive.

### E.3 `torchdde`'s memory and time benchmark

In this subsection, we provide time and memory benchmarks on some of `torchdde`'s solvers. In order to compare both training methods of optimize-then-discretize (i.e. the adjoint method) and discretize-then-optimize (i.e. regular backpropagation), we present the Brusselator's experiment time duration and memory usage for various solvers during training (with a batch size of 1024) in Tables 4 and 5, respectively. The results are as expected : the adjoint method is slower (by a small factor) and consumes less memory than the regular backpropagation. These results are consistent with NODE's examination of the adjoint method and conventional backpropagation tradeoffs Chen et al. (2018).

|       | Adjoint          | Backpropagation   |
|-------|------------------|-------------------|
| RK4   | $4.8 \pm 0.23$   | $1.89 \pm 0.09$   |
| RK2   | $2.4 \pm 0.005$  | $0.90 \pm 0.005$  |
| Euler | $1.5 \pm 0.01$   | $0.47 \pm 0.003$  |

Table 4: Clock Time (s) per batch

|       | Adjoint          | Backpropagation   |
|-------|------------------|-------------------|
| RK4   | $2.2 \pm 18$     | $2.87 \pm 4$      |
| RK2   | $2.15 \pm 20$    | $2.48 \pm 3$      |
| Euler | $2.09 \pm 15$    | $2.264 \pm 9$     |

Table 5: GPU consumption (Gb $\pm$ Mb) per batch

Figure 14 & 15 compares respectively time and memory consumption for a forward of a Neural DDE with varying number of delays, each having approximately 28k parameters. We use the following setup :

- Use a RK4 solver with a timestep of $dt = 0.1$.

- Use a batch size of 128.

- Neural DDE uses [1, 3, 5, 10, 20] delays.

- State $g(t)$ dimension is 100 .

- The upper integration bound is varied from $t = 0.2$ to $t = 5.0$s.

- The notation "#i" in the figure refers to the number of delays used in the Neural DDE.

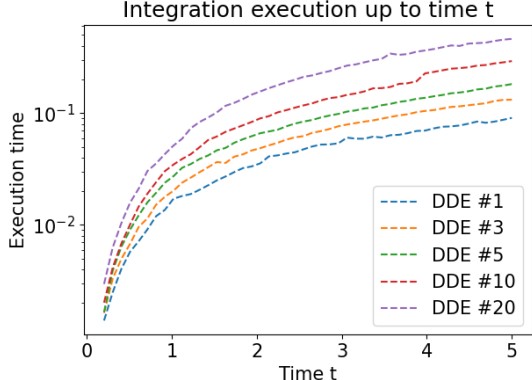

Figure 14: Time duration of forward pass averaged over 5 runs

Figure 15: Memory consumption of forward pass averaged over 5 runs

## F Additional experiments

We present the Shallow Water equation dataset, available in the PDEBenchmark suite (Takamoto et al., 2023). We put ourselves in the highly restrictive partially-observable setting by randomly sampling 4 points on the spatial grid and fit its dynamics. Finally, Table 6 presents a summary of the test loss from this experiments along with the number of parameters for each model in Table 7. Figure 16 illustrates the performance of each model on the Shallow Water testset. Excluding NODE, all models yield satisfactory outcomes. However, when evaluating based on Mean Squared Error (MSE), NDDEs yielded better performance than the presented baselines.

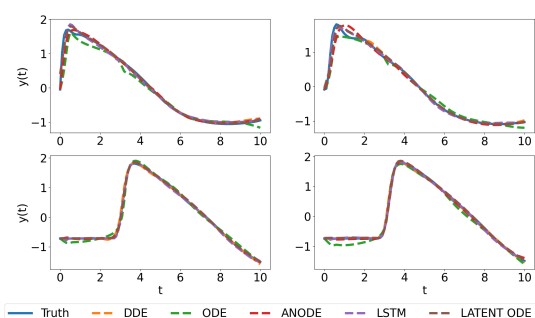

Figure 16: Random test sampled of Shallow Water dataset

|  | LSTM | NODE | ANODE | Latent ODE | NDDE |
|---|---|---|---|---|---|
| Shallow Water | $0.0031 \pm 0.001$ | $0.046 \pm 0.035$ | $0.004 \pm 0.001$ | $0.0058 \pm 0.001$ | $\mathbf{0.001 \pm 0.0001}$ |

Table 6: Shallow Water test loss experiments averaged over 5 runs

| | LSTM | NODE | ANODE | Latent ODE | NDDE |
|---|---|---|---|---|---|
| Shallow Water | 2512 | 2404 | 2534 | 5853 | 2662 |

Table 7: Number of parameters for Shallow Water experiment

# G    Proof of Theorem 3.1

Proof is inspired from Calver & Enright (2017) and put into ML context.
For conciseness, we use the following notation $x(t) = x_t$, $\lambda(t) = \lambda_t$, etc...

We want to solve the optimization problem where $\tau$ may appear in our parameter vector $\theta$ :

$$\arg\min_{\theta} \ L(x_t),$$

$$s.t. \ L(x_t) = \int_0^T l(x_t)dt,$$

$$\dot{x}_t - f(x_t, x_{t-\tau}; \theta) = 0,$$

$$x_{t \leq 0} = \psi(t). \tag{26}$$

We consider the following Lagrangian :

$$J = L + \int_0^T \lambda_t\big(\dot{x}_t - f(x_t, x_{t-\tau}; \theta)\big)dt.$$

$$\implies \frac{\mathrm{d}J}{\mathrm{d}\theta} = \frac{\mathrm{d}L}{\mathrm{d}\theta} \tag{27}$$

Integration by parts yields :

$$J = \int_0^T l(x_t)dt + \Big[\lambda_t x_t\Big]_0^T - \int_0^T \dot{\lambda}_t x_t + \lambda_t f(x_t, x_{t-\tau}; \theta)dt. \tag{28}$$

Taking the derivative w.r.t. $\theta$ :

$$\frac{\mathrm{d}J}{\mathrm{d}\theta} = \int_0^T \frac{\partial l(x_t)}{\partial x_t}\frac{\partial x_t}{\partial \theta}dt + \lambda_T \frac{\mathrm{d}x_T}{\mathrm{d}\theta} - \lambda_0 \overset{0}{\cancel{\frac{\mathrm{d}x_0}{\mathrm{d}\theta}}}$$

$$+ \int_0^T -\dot{\lambda}_t \frac{\partial x_t}{\partial \theta}dt + \int_0^T -\lambda_t \frac{\partial f(x_t, x_{t-\tau}; \theta)}{\partial \theta}dt$$

$$+ \int_0^T -\lambda_t \frac{\partial f(x_t, x_{t-\tau}; \theta)}{\partial x_t}\frac{\partial x_t}{\partial \theta}dt + \int_0^T -\lambda_t \frac{\partial f(x_t, x_{t-\tau}; \theta)}{\partial x_{t-\tau}}[\frac{\partial x_{t-\tau}}{\partial \theta} + x'_{t-\tau}\frac{\partial t - \tau}{\partial \theta}]dt. \tag{29}$$

Since $\frac{\partial t-\tau}{\partial \theta} = \frac{\partial t-\tau}{\partial \tau} = -1$ the following equation simplifies to :

$$\frac{\mathrm{d}J}{\mathrm{d}\theta} = \int_0^T \frac{\partial l(x_t)}{\partial x_t}\frac{\partial x_t}{\partial \theta}dt + \lambda_T \frac{\mathrm{d}x_T}{\mathrm{d}\theta} - \lambda_0 \overset{0}{\cancel{\frac{\mathrm{d}x_0}{\mathrm{d}\theta}}}$$

$$+ \int_0^T -\dot{\lambda}_t \frac{\partial x_t}{\partial \theta}dt + \int_0^T -\lambda_t \frac{\partial f(x_t, x_{t-\tau}; \theta)}{\partial \theta}dt$$

$$+ \int_0^T -\lambda_t \frac{\partial f(x_t, x_{t-\tau}; \theta)}{\partial x_t}\frac{\partial x_t}{\partial \theta}dt + \int_0^T \lambda_t \frac{\partial f(x_t, x_{t-\tau}; \theta)}{\partial x_{t-\tau}}\left[\frac{\partial x_{t-\tau}}{\partial \theta} + x'_{t-\tau}\right]dt. \tag{30}$$

**Rearranging integrals**

We rework the first part of the last term to close the equation on $x_t$ :

$$\int_0^T \lambda_t \frac{\partial f(x_t, x_{t-\tau}; \theta)}{\partial x_{t-\tau}} \frac{\partial x_{t-\tau}}{\partial \theta} dt = \int_{-\tau}^{T-\tau} -\lambda_{t+\tau} \frac{\partial f(x_{t+\tau}, x_t; \theta)}{\partial x_t} \frac{\partial x_t}{\partial \theta} dt. \tag{31}$$

$$\int_0^T -\lambda_t \frac{\partial f(x_t, x_{t-\tau}; \theta)}{\partial x_{t-\tau}} \frac{\partial x_{t-\tau}}{\partial \theta} dt = \int_{-\tau}^{T-\tau} -\lambda_{t+\tau} \frac{\partial f(x_{t+\tau}, x_t; \theta)}{\partial x_t} \frac{\partial x_t}{\partial \theta} dt. \tag{32}$$

Chosing the multipliers so that $\lambda_{t \geq T} = 0$, we get :

$$\int_0^T -\lambda_t \frac{\partial f(x_t, x_{t-\tau}; \theta)}{\partial x_{t-\tau}} \frac{\partial x_{t-\tau}}{\partial \theta} dt = \int_{-\tau}^{T} -\lambda_{t+\tau} \frac{\partial f(x_{t+\tau}, x_t; \theta)}{\partial x_t} \frac{\partial x_t}{\partial \theta} dt. \tag{33}$$

Finally, observing that $\frac{\partial x_{t \leq 0}}{\partial \theta} = 0$, the last term becomes :

$$\int_0^T -\lambda_t \frac{\partial f(x_t, x_{t-\tau}; \theta)}{\partial x_{t-\tau}} \frac{\partial x_{t-\tau}}{\partial \theta} dt = \int_0^{T} -\lambda_{t+\tau} \frac{\partial f(x_{t+\tau}, x_t; \theta)}{\partial x_t} \frac{\partial x_t}{\partial \theta} dt. \tag{34}$$

**Adjoint Equation**

Finally, injecting this result, we rearrange the terms in Eq.29 :

$$\begin{aligned}
\frac{dJ}{d\theta} = &-\int_0^T \left( \dot{\lambda}_t - \frac{\partial l(x_t)}{\partial x_t} + \lambda_t \frac{\partial f(x_t, x_{t-\tau}; \theta)}{\partial x_t} + \lambda_{t+\tau} \frac{\partial f(x_{t+\tau}, x_t; \theta)}{\partial x_t} \right) \frac{\partial x_t}{\partial \theta} dt \\
&-\int_0^T \lambda_t \left( \frac{\partial f(x_t, x_{t-\tau}; \theta)}{\partial \theta} + \frac{\partial f(x_t, x_{t-\tau}; \theta)}{\partial x_{t-\tau}} x'_{t-\tau} \right) dt + \lambda_T \frac{dx_T}{d\theta} \!\!\!\!\nearrow^0 \\
&+\int_{-\tau}^0 \lambda_{t+\tau} \frac{\partial f(x_{t+\tau}, x_t; \theta)}{\partial x_t} \frac{\partial \psi_t}{\partial \theta} dt.
\end{aligned} \tag{35}$$

The last term vanishes because of the chosen adjoint final condition $\lambda_{t \geq T} = 0$, thus we get the following adjoint dynamics, to be integrated backwards in time :

$$\dot{\lambda}_t = \frac{\partial l(x_t)}{\partial x_t} - \lambda_t \frac{\partial f(x_t, x_{t-\tau}; \theta)}{\partial x_t} - \lambda_{t+\tau} \frac{\partial f(x_{t+\tau}, x_t; \theta)}{\partial x_t}, \tag{36}$$

$$\lambda_{t \geq T} = 0. \tag{37}$$

Hence, the gradient's loss w.r.t to the parameters is :

$$\begin{aligned}
\frac{dJ}{d\theta} = &-\int_0^T \lambda_t \left( \frac{\partial f(x_t, x_{t-\tau}; \theta)}{\partial \theta} + \frac{\partial f(x_t, x_{t-\tau}; \theta)}{\partial x_{t-\tau}} x'_{t-\tau} \right) dt + \\
&+\int_{-\tau}^0 \lambda_{t+\tau} \frac{\partial f(x_{t+\tau}, x_t; \theta)}{\partial x_t} \frac{\partial \psi_t}{\partial \theta} dt.
\end{aligned} \tag{38}$$

The last term is more often than null since the history function is parameter independent simplifying ever further the equation.

**Notes on the derivative of the loss**

Practically, the loss $L(x_t)$ is evaluated from a finite number $N$ of points in time :

$$L(x_t) = \int_0^T l(x_t)dt \tag{39}$$

$$= \int_0^T \Big[ \sum_{i=1}^N \bar{l}(x_{t_i})\delta(t - t_i)\Big]dt. \tag{40}$$

With $\bar{l}$ a function computing the objective for each sampled point. This yields the following gradient :

$$\frac{\partial l(x_t)}{\partial x_t} = \sum_{i=1}^N \frac{\partial \bar{l}(x_{t_i})}{\partial x_{t_i}}\delta(t - t_i). \tag{41}$$

This term is then always null, except for $t = t_i$, this is why the adjoint dynamics in Eq.*equation* 35 are integrated from one sampling point $t_i$ to the previous $t_{i-1}$, where the adjoint state is incremented as follows :

$$\lambda_{t_i^-} = \lambda_{t_i^+} - \frac{\partial \bar{l}(x_{t_i})}{\partial x_{t_i}}. \tag{42}$$

which corresponds to integrating the dirac in Eq.equation 41 in reverse time for an infinitesimal time.

**Case for multiple constant delays**

For the case of multiple delays $\tau_i$, the first term $\lambda_t \frac{\partial f(x_t, x_{t-\tau};\theta)}{\partial x_t}$ in the adjoint dynamics (Eq. 36) is replaced by :

$$\lambda_t \frac{\partial f(x_t, x_{t-\tau_1}, \ldots, x_{t-\tau_k};\theta)}{\partial x_t} \tag{43}$$

and the term $\lambda_{t+\tau}\frac{\partial f(x_{t+\tau}, x_t)}{\partial x_{t+\tau}}$ in the adjoint dynamics (Eq. 36) is replaced by the following :

$$\sum_{i=0}^n \lambda_{t+\tau_i} \frac{\partial f(x_{t+\tau_i}, x_{t-\tau_0+\tau_i}, \ldots, x_{t-\tau_n+\tau_i})}{\partial x_{t+\tau_i}} \tag{44}$$

For the gradient $\frac{\partial f(x_t, x_{t-\tau};\theta)}{\partial x_{t-\tau}}x'_{t-\tau}$ in Equation 38 is replaced by the following :

$$\sum_{i=0}^n \frac{\partial f(x_t, x_{t-\tau_0}, \ldots, x_{t-\tau_n})}{\partial x_{t-\tau_i}}x'_{t-\tau_i} \tag{45}$$

Finally $\int_{-\tau}^0 \lambda_{t+\tau}\frac{\partial f(x_{t+\tau}, x_t;\theta)}{\partial x_t}\frac{\partial \psi_t}{\partial \theta}$ from Equation 38 is replaced by :

$$\sum_{i=0}^n \int_{-\tau_i}^0 \lambda_{t+\tau_i} \frac{\partial f(x_{t+\tau_i}, x_{t-\tau_0+\tau_i}, \ldots, x_{t-\tau_n+\tau_i})}{\partial x_t}\frac{\partial \psi_t}{\partial \theta} \tag{46}$$

## H    Training Neural DDE algorithm with several delays

In this Appendix, we present the equivalent of Algorithm 1 for multiple constant delayed Neural DDEs, highlighting several key differences. First, it is necessary to allocate $\tau_{max}$ of the training trajectories to compute the history function of the DDE. Additionally, the adjoint dynamics and the gradient computation become more complex due to the presence of multiple delays.

---

**Algorithm 2** Training a Neural DDE with learnable delays with the adjoint method.

---

**Require:** Dataset of one trajectory $\mathcal{D} = \{(t_0, x_0), \ldots, (t_N, x_N)\}$.
**Require:** Initialized model $f_\theta$.
**Require:** Initialized $k$ positive delays $\tau_1, \ldots, \tau_k$ that can appear in the parameters vector $\theta$.
  1: **for** $i \leftarrow 1, \ldots, N_{epochs}$ **do**
  2:     Set $\tau_{max} = \max(\tau_1, \ldots, \tau_k)$
  3:     Create history function interpolation $\psi$ with data from $\mathcal{D}$ such that $t < \tau_{max}$.
  4:     Solve DDE dynamics:
  5: $\begin{cases} x'(t) = f_\theta(t, x(t), x(t - \tau_1), \ldots, x(t - \tau_k)) \\ x(t < \tau_{max}) = \psi(t) \end{cases}$ .
  6:     Compute loss $L(x(t_N)) = \int_{\tau_{max}}^{t_N} l(x(s))\, ds$
  7:     Solve Adjoint dynamics :
  8: $\begin{cases} \lambda'(t) = \frac{\partial l(x(t))}{\partial x(t)} - \lambda(t) \frac{\partial f_\theta(x(t), x(t-\tau_1), \ldots, x(t-\tau_k))}{\partial x(t)} - \sum_{i=1}^{k} \lambda(t + \tau_i) \frac{\partial f_\theta(x(t+\tau_i), x(t-\tau_0+\tau_i), \ldots, x(t-\tau_n+\tau_i))}{\partial x(t+\tau_i)} \\ \lambda(t \geq t_N) = 0. \end{cases}$ .
  9:     Compute $\frac{dL}{d\theta}$ :
 10:

$$\begin{aligned} \frac{dL}{d\theta} = &-\int_0^T \lambda(t) \left( \frac{\partial f_\theta(x(t), x(t-\tau))}{\partial \theta} + \sum_{i=0}^{n} \frac{\partial f_\theta(x(t), x(t-\tau_0), \ldots, x(t-\tau_n))}{\partial x(t-\tau_i)} x'(t-\tau_i) \right) dt \\ &+ \sum_{i=0}^{n} \int_{-\tau_i}^{0} \lambda(t+\tau_i) \frac{\partial f(x(t+\tau_i), x(t-\tau_0+\tau_i), \ldots, x(t-\tau_n+\tau_i))}{\partial x(t)} \frac{\partial \psi(t)}{\partial \theta} \end{aligned} \tag{47}$$

 11:
 12:     Update $\theta$
 13: **end for**

---

## I    Proof of Proposition 2.2

Let us start by stating Takens' theorem as expressed by (Noakes, 1991; Takens, 1981):

**Theorem I.1** (Takens' embedding theorem)**.** *Let $M$ be compact. There is an open dense subset $\mathcal{D}$ of $Diff(M) \times C^k(M, \mathbb{R})$ with the property that the Takens map*

$$h : M \rightarrow \mathbb{R}^{2m+1}$$

*given by $h(x) = (g(x), g(\phi(x)), g(\phi \circ \phi(x)), \ldots g(\phi^{2m}(x)))$ is an embedding of $C^k$ manifolds, when $(\phi, g) \in \mathcal{D}$.*

Here, $\phi$ stands for the operator that advances the dynamical system by a time step $\tau$, i.e. that sends $x(t)$ to $x(t+\tau)$, and $g$ is the observable operator, that sends a full state $x(t)$ to actual observables $g(x(t)) =: g(t)$. Variants of this Theorem, e.g. Sauer et al. (1991), include the consideration of any set of different delays $\tau_i$ instead of uniformly spaces ones. The representation $h(x(t)) = (g(x(t)), g(x(t - \tau)), g(x(t - 2\tau)), \ldots g(x(t - 2m\tau))$ then becomes $h(x(t)) = (g(x(t)), g(x(t - \tau_1)), g(x(t - \tau_2)), \ldots g(x(t - \tau_{2m}))$. In the proof of Takens' theorem, $m$ is the intrinsic dimension of the dynamical system, i.e. the one of the manifold $M$.

Now, given the full state $x$ that follows the dynamics :

$$\frac{\mathrm{d}x}{\mathrm{d}t} = G(x), \quad x(0) = x_0 \tag{48}$$

we use the chain rule on the observable $g$:

$$\frac{\mathrm{d}g}{\mathrm{d}t} = g'(x(t))\, G(x) \tag{49}$$

By applying the inverse of the delay coordinate map $h^{-1}$ from theorem I.1, which is invertible from its image as it is an embedding, we show that $g$'s dynamics possesses a DDE structure:

$$\frac{\mathrm{d}g}{\mathrm{d}t} = (g' \times G) \circ h^{-1}(g(t), g(t - \tau_1), \dots, g(t - \tau_n)) \tag{50}$$

The proof is completed by choosing $f = (g' \times G) \circ h^{-1} - M$ where $M$ is obtained by the Mori-Zwanzig formalism (Equation (5)).

Note that to be able to apply Takens' theorem, we needed the step-forward operator $\phi$ to be a diffeomorphism, i.e. the flow of the dynamical system to be smooth and smoothly invertible. Also, we used the differentiability of the observables $g$ to express $g'$.

## J  The importance of relevant delays

Let us consider a dynamical system evolving on a compact smooth manifold $\mathcal{S} \subset \mathbb{R}^d$, assumed to be an attractor. Let us consider a $C^2$ observable function $g : \mathcal{S} \to \mathbb{R}$.

Takens' theorem (Takens, 1981) rigorously discusses conditions under which a delay vector of a scalar-valued observable $(g(x(t)), g(x(t - \tau)), \dots, g(x(t - p\tau)))$, $p \in \mathbb{N}$ defines an embedding, a smooth diffeomorphism onto its image. It guarantees a topological equivalence between the original dynamical system and the one constructed from the memory of the observable. The dynamics of the system can then be reformulated on the set $(g(x(t)), g(x(t - \tau)), \dots, g(x(t - p\tau)))$.

The Takens' theorem, later extended by Sauer et al. (1991), establishes a sufficient condition but does not provide information about the time delay $\tau$. From a mathematical viewpoint, the delay could be arbitrary, besides some pointwise values excluded by the theorem. In practice however, its value is instrumental in a successful embedding. If too small, entries of the delay vector data are too similar; if too large, the entries tend to be completely uncorrelated and cannot be numerically linked to a consistent dynamical system.

We here illustrate the impact of suitable delays in the relevance of the information available to inform the future evolution of the observable. We consider a simple 2-delay dynamical system described by:

$$
\begin{aligned}
g(t + \Delta t) &= \cos(g(t - \tau_1))\, \sin(g(t - \tau_2)) - \alpha \operatorname{sinc}(3\, g(t - \tau_1)) + \alpha\, \cos(g(t - \tau_2)), \\
g(t < 0) &= \psi(t),
\end{aligned}
$$

with $\alpha = 0.2$, $\tau_1 = p_1^\star \Delta t$, $\tau_2 = p_2^\star \Delta t$, $p_1^\star = 125$, $p_2^\star = 200$.

The relevance of the delays $\{\tau_1, \tau_2\}$ for informing $g(t + \Delta t)$ is assessed in terms of the mutual information $I\left((g(t - \tau_1), g(t - \tau_2)), g(t)\right)$ and shown in Fig. 17 as a 2-D map in terms of $p_1$ and $p_2$. The map is symmetric, consistently with the symmetry of the mutual information, $I\left((g(t - \tau_1), g(t - \tau_2)), g(t)\right) = I\left((g(t - \tau_2), g(t - \tau_1)), g(t)\right)$.

It can be seen that the amount of information shared between the current observation and a delay vector of the observable widely varies with the delays. The ability of the present Neural DDE method to learn the delays, in addition to the model $f_\theta$, is thus key to its performance and wide applicability.

# K   Training hyperparameters

Our training approach incorporates a progressive strategy considered to be curriculum learning strategy Soviany et al. (2022). We begin by feeding the models shorter trajectory segments and gradually increase their length when the patience hyperparameter is exceeded. This process continues until we reach the desired trajectory length. Each time the trajectory length is increased, we reset the patience hyperparameter to 0. It is then incremented by 1 if the validation loss fails to decrease, and reset to 0 if the validation loss improves. This method aligns with the principles of curriculum learning, a technique that involves training machine learning models in a structured order, typically progressing from simpler to more complex examples. In our case, this translates to moving from shorter to longer trajectories. This approach aims to enhance the learning process and improve model performance. Table 8 displays the patience hyperparameter and how much trajectory length was given initially. Table 9 refers to the number of parameters of each model. The loss function used across all experiments is the MSE loss, and we employ the Adam optimizer with a weight decay of $10^{-7}$. Table 14 provides the initial and final learning rates $(lr_i, lr_f)$ for each experiment, which are associated with the scheduler. The scheduler is a StepLR scheduler with a gamma factor $(\gamma = \exp\left\{\frac{\log \frac{lr_f}{lr_i}}{N}\right\}$, where $N$ is the trajectory's length). The scheduler adjusts the learning rate as the trajectory length increases, allowing training to start with the initial learning rate $lr_i$ and gradually decrease to the final learning rate $lr_f$. All continuous-time models (NODE, ANODE, Latent ODE and NDDE) used RK4 for numerical integration. Table 11 shows the width and depth of the MLPs for NODE, ANODE, and NDDE across all experiments. Additionally, we provide the hidden size and number of layers for the LSTM model in Table 13. Finally, Table 12 summarizes the Latent ODE hyperparameters, where the vector field $f_\theta$ (defined in the introduction) is an MLP with the width and depth specified in the second and third columns, the latent size of $z_0$ in the last column, and the RNN's hidden size in the fourth column. If some models has less parameters compared to others it is that we found that they provided better results with less. ANODE's augmented state dimension matches that of the number of delays used by NDDE displayed in Table 1. Due to the inherent different nature of each model, they provide output of different length and don't necessarily start at the same initial time $t_0$ : Table 10 provides a MSE comparison along the common trajectory predicted by all models. Compared to Table 2, the results barely differ.

|  | KS | Cavity | Brusselator |
| --- | --- | --- | --- |
| Length Start | 15% | 50% | 25% |
| Patience | 40 | 50 | 20 |

Table 8: How long is the trajectory chunks given at first and the patience used for each experiment

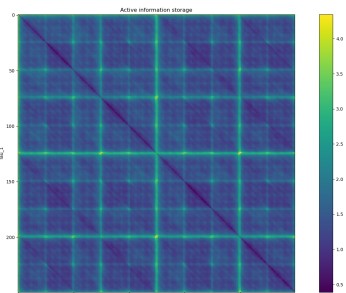

Figure 17: $\{\tau_1 = p_1 \, \Delta t, \tau_2 = p_2 \, \Delta t\}$-map of Delayed Mutual Information, $I\left((g(t - \tau_1), g(t - \tau_2)), g(t)\right)$. The maximum is exhibited at $(125, 200)$ and $(200, 125)$, in accordance with $p_1^\star = 125$, $p_2^\star = 200$.

|  | LSTM | NODE | ANODE | Latent ODE | NDDE |
|---|---|---|---|---|---|
| Brusselator | 1764 | 3265 | 3395 | 3666 | 3331 |
| KS | 18130 | 9029 | 11609 | 8118 | 19343 |
| Cavity | 2234 | 2209 | 2274 | 3642 | 2242 |

Table 9: Number of parameters for each experiment

|  | LSTM | NODE | ANODE | Latent ODE | NDDE |
|---|---|---|---|---|---|
| Brusselator | **0.0051 ± 0.0031** | 0.75 ± 0.0014 | **0.0050 ± 0.0050** | 0.014 ± 0.0076 | 0.011 ± 0.0076 |
| KS | 0.77 ± 0.061 | 0.71 ± 0.10 | 0.53 ± 0.052 | 0.43 ± 0.07 | **0.30 ± 0.032** |
| Cavity | 0.75 ± 0.51 | 0.96 ± 0.0001 | 0.65 ± 0.021 | 0.25 ± 0.14 | **0.13 ± 0.0081** |

Table 10: Test loss experiments averaged over 5 runs over common trajectory predictions

|  | NODE/ANODE/NDDE | |
|---|---|---|
|  | Width | Depth |
| Brusselator | 32 | 4 |
| KS | 64 | 3 |
| Cavity | 32 | 3 |
| Shallow Water | 32 | 3 |

Table 11: MLP width and depth for each experiment

| Experiment | Width Size | Depth | Hidden Size | Latent Size |
|---|---|---|---|---|
| Brusselator | 16 | 3 | 16 | 16 |
| KS | 32 | 3 | 16 | 16 |
| Cavity | 16 | 3 | 8 | 8 |
| Shallow Water | 32 | 3 | 8 | 8 |

Table 12: Configuration parameters for each experiment for Latent ODE

| Experiment | Hidden Size | Number of Layers |
|---|---|---|
| Brusselator | 5 | 10 |
| KS | 25 | 5 |
| Shallow Water | 6 | 10 |
| Cavity | 7 | 7 |

Table 13: Hidden size and number of layers for each experiment for LSTM model

| Experiment | $lr_i$ | $lr_f$ |
|---|---|---|
| Brusselator | 0.001 | 0.0001 |
| Cavity | 0.005 | 0.00005 |
| KS | 0.01 | 0.0001 |
| Shallow Water | 0.001 | 0.00001 |

Table 14: Initial and final learning rates for each experiment

