# OpenReview forum: "Neural DDEs with Learnable Delays for Partially Observed Dynamical Systems"
_TMLR — Rejected by TMLR_

### Review · Reviewer_JLGw · 2024-10-17

**Summary Of Contributions:**

The paper introduces neural delay differential equations (NDDEs) with learnable delays for modeling partially observed dynamical systems. It provides a theoretical foundation using the Mori-Zwanzig formalism and Takens' embedding theorem. The method is tested on various systems, including synthetic, chaotic, and real-world fluid dynamics data, and it is compared favorably against baseline models like LSTM, NODE, ANODE, and latent ODE. The authors make their code available, promoting reproducibility.

**Audience:**

Yes

**Claims And Evidence:**

Yes

**Requested Changes:**

1. theoretical clarifications:
   1.1 add intuitive explanations and visualizations for theoretical parts (recommended)
   1.2 clarify conditions for takens' theorem to apply and ensure they are met (crucial)
   1.3 provide clearer distinction between memory and state in different models (RNNs, augmented NODEs, NDDEs) (recommended)
   1.4 explain connections between non-markovian systems and partial observability more clearly (recommended)
   1.5 clarify the contribution compared to Oprea et al. 2023 and Schlaginhaufen et al. 2021. (crucial)

2. experimental improvements:
   2.1 include ablation study with fixed vs. learned delays (crucial)
   2.2 add experiments with higher-dimensional systems (e.g., lorenz-96 with varying dimensions) (recommended)
   2.3 provide more details on impact of different initialization for learned delay choices (recommended)
   2.4 include additional performance metrics beyond MSE (e.g., power spectrum correlation, lyapunov exponents, or state space reconstruction metrics) (recommended)
   2.5 conduct a systematic study of performance under different noise levels (optional)
   2.6 analyze long-term behavior of learned dynamics (recommended)
   2.7 show application to more real-world applications, e.g, in climate science or biology (optional)
   2.8 add baseline of reservoir computing (recommended)
   2.9 add baseline of other NDDE approaches (especially, Oprea et al. 2023 and maybe also Schlaginhaufen et al. 2021) (crucial)

3. discussion enhancements:
   3.1 add in-depth discussion of potential limitations and challenging scenarios (crucial)
   3.2 include more clear quantitative comparison of computational aspects (optional)
   3.3 add section on future directions (optional)

4. related work:
   4.1 include discussion of reservoir computing for partially observed systems (crucial)
   4.2 mention recent advances in koopman operator theory (optional)

5. presentation improvements:
   5.1 add graphical abstract or high-level diagram of the main concept (recommended)
   5.2 include algorithm or pseudocode for NDDE (optional)
   5.3 improve visual hierarchy (section headings, subsection headings, paragraph breaks) (recommended)
   5.4 enhance table and figure presentations (larger labels, more context in captions, use vector graphics) (crucial)

6. language and structure:
   6.1 improve conciseness and clarity throughout the paper (crucial)
   6.2 use grammar checker to correct errors (crucial)
   6.3 avoid informal language and passive voice (recommended)
   6.4 restructure introduction to separate related work more clearly (recommended)

**Strengths And Weaknesses:**

Strengths:
- solid theoretical grounding for using NDDEs to model partially observed systems, leveraging the Mori-Zwanzig formalism (justified by Takens' embedding theorem)
- use of learnable delays in NDDEs is an good idea (seems to have been proposed by Oprea et al. 2023) that addresses limitations of existing methods for modeling partially observed systems
- variety of systems, including synthetic data, chaotic systems, and real-world fluid dynamics data
- compares NDDEs against several baseline models (LSTM, NODE, ANODE, Latent ODE)
- made their code available, which promotes reproducibility

Weaknesses:

Theoretical and Methodological Issues:
- clarity: particularly the theoretical parts, could benefit from more intuitive explanations or visual aids to help readers grasp the concepts more easily  → add intuitive explanations or visualizations
- could clarify conditions for Taken's theorem to apply and to make sure they are met (and inform the reader when to be careful with applying this method)
- I find the paper a bit vague in the distinction between memory and state, please clarify: in RNNs, memory is encoded in hidden states, (i.e. learned representations), in Augmented NODEs, additional dimensions act as a form of memory, in NDDEs, past states serve as explicit memory. these are contrasted, but the comparison of strengths and limitations of these is a bit vague
- the connections between Non-Markovian systems to partial observability (via Taken's theorem) could be made clearer

Experimental Design and Analysis:
- more details on impact of different initial delay choices
- ablation study showing performance with fixed vs. learned delays seems missing
- experiments with higher-dimensional systems (e.g., Lorenz-96 with varying dimensions) would demonstrate the method's scalability
- the comparison metric is MSE which is fine, but not very comprehensive and (an aspect that is rarely mentioned in the literature) maybe we are not only interested in predictive performance but also in capturing the true underlying dynamics  → a concrete question I have is: how does the long-term behavior of the learned dynamics look like, do we get chaotic dynamics if we let the system run freely for long periods of time or does it break down after a while and reach fixed point or limit cycle dynamics. the paper shows some long-term predictions, it doesn't address the stability of these predictions over very long time horizons
- if using only MSE, then at least better justification of that decision would be warranted

Limitations and Future Work:
- more in-depth discussion of potential limitations or scenarios where the method might not perform well would balance it, potential limitations I could think of are
	- computational complexity as the number of delays increases
	- challenges in determining the optimal number of delays
	- potential problems in high-dimensional delay spaces, or in very noisy settings
	- limitations in capturing very long-term dependencies
- there's some discussion of computation, but more clear quantitative comparison would help
- could add a few sentences on future directions

Related Work:
- why not include reservoir computing for partially observed systems as related work? seems to be still close to state-of-the-art and is fast and easy to set up and train
- in terms of related work it may be also good to mention recent advances in Koopman operator theory
- why not include other NDDE approaches as comparison (Oprea, Schlaginhaufen)? What is precisely the difference to Oprea et al. 2023?

Additional Suggestions:
- more real-world applications as in climate science or biology would benefit the paper
- systematic study of performance under different noise levels would be valuable
- there are geometric measures that could be helpful for analysing long-term dynamical behavior, e.g., Koppe et al. 2019 "Identifying nonlinear dynamical systems via generative recurrent neural networks with applications to fMRI"
- but more generally, more metrics would paint a more complete picture - e.g., power spectrum correlation (especially for periodic systems), comparing lyapunov exponents (for chaotic) (in general an analysis of the prediction horizon would be interesting), or state space reconstruction metrics as the geometric measure mentioned above

Minor Issues:
- RNNs are critiqued as being discrete, but approximation used here is also discrete (or did I misunderstand?)

Presentation:
- clarity: while for the target audience this should be fine, often a brief explanation of technical terms would make the paper more readable, e.g., taken's delay embedding theorem, MZ formalism etc.
- graphical abstract or high-level diagram of the main concept (NDDEs for partially observed systems) would make it much easier to understand quickly what the paper is about
- algorithm or pseudocode for NDDE may make the method more clear
- visual hierarchy in terms of section headings, subsection headings, and paragraph breaks could be more clearly formatted
- table 2: add "model performance (MSE)" or similar instead of just model names
- figure 1: captions are quite small, also figure 4, 5 the labels should be larger
- figure 3: add more context about what this represents
- some figures are pixely (minor critique) images, better with pdf/svg instead of png/jpg

Language:
- language could be improved in many places to make it more concise and clear:
	- e.g., "For example, an instance of this issue" → wordy, and doubling "example" = "instance"
	- "this brings us to the domain of"
	- there's many more such places of wordiness, a simple grammar checker would help already a lot here
- some errors: e.g.: "NDDEs integration *are*  at least an order of magnitude faster"  → is at least ... ⇒ use a grammer checker
- informal language "In a nutshell, the Brusselator and KS System experiments"
- passive voice "The system is integrated over" → We integrate
-  "Utilizing the Mori-Zwanzig (MZ) formalism from statistical physics, we demonstrate that Constant Lag Neural Delay Differential Equations (NDDEs) naturally serve as suitable models for partially observed states." → split in two sentences
overall clarity is a bit obscured by the language and the structure is not flowing not entirely coherent, e.g. the introduction is mixed with a related work section that could be more clearly broken into paragraphs with headings

Questions:
- how sensitive is the NDDE performance to the choice of hyperparameters, particularly the number of delays? how to choose it?   → sensitivity analysis for key hyperparameters would be helpful
- it's not clear to me how learning the delays works, how is this done in practice?
- are there any numerical stability issues that can arise when solving delay differential equations, especiallly for chaotic systems like KS?
- setup for partial observability for KS is not clear to me: how are the observed variables are chosen and in general how does this choice impact the results?

---

> ### Author Response · Authors · 2024-11-13
>
> > clarity: particularly the theoretical parts, could benefit from more intuitive explanations or visual aids to help readers grasp the concepts more easily → add intuitive explanations or visualizations
>
> We provide enhancements to the introduction and Figure 1. We hope the modifications in the introduction and figure add more context and elucidates the reviewer's questions.
>
> > could clarify conditions for Taken's theorem to apply and to make sure they are met (and inform the reader when to be careful with applying this method)
>
> We changed the theorem 2.1 to not include any reference to Appendix B and provided a discussion on how to deal with the noise term $F$ that refers to Appendix B. Proposition 2.2 was also updated to give the explicit conditions for the dynamical system and the observable $g$.
>
> > I find the paper a bit vague in the distinction between memory and state, please clarify: in RNNs, memory is encoded in hidden states, (i.e. learned representations), in Augmented NODEs, additional dimensions act as a form of memory, in NDDEs, past states serve as explicit memory. these are contrasted, but the comparison of strengths and limitations of these is a bit vague.
>
> We discuss how some models incorporate "memory". This is done via the addition of extra variables. Those extra variables can be latent variables (like in the LSTM), past states (like in the NDDE) or augmented variables (like in ANODE). DDEs have the additional benefit of a clearer interpretation with an explicit memory formulation.
>
> Strengths and limitations : RNNs have the appealing ability to compress past states into a hidden latent variable. However, they suffer from some issue like its exploding/vanishing gradient problem (https://arxiv.org/pdf/2405.21064) Limitations of Neural ODE is that it is Markovian. ANODE with its augmented variable $a(t)$ can lack interpretability.
>
> > the connections between Non-Markovian systems to partial observability (via Taken's theorem) could be made clearer.
>
> Proposition 2.2 connects partially observed systems with DDEs. Partially observed systems cannot be described by a Markovian model since observables are not conjugate to a state vector. Instead, they are described by a non-Markovian formulation, such as DDEs.
>
> > more details on impact of different initial delay choices
> > ablation study showing performance with fixed vs. learned delays seems missing
>
> Figure 10 provided an ablation study for fixed and learnt delays. We decided to improve this experiment by experimenting with different delay initialization values and added our insights in the revised paper in Section 4.2 in the cavity paragraph.
>
> > experiments with higher-dimensional systems (e.g., Lorenz-96 with varying dimensions) would demonstrate the method's scalability.
>
> Experiments involving very large systems are left for future work. However, the method seems to scale since we seamlessly trained networks with 8M parameters with the present experiments.
>
> > the comparison metric is MSE which is fine, but not very comprehensive and (an aspect that is rarely mentioned in the literature) maybe we are not only interested in predictive performance but also in capturing the true underlying dynamics → a concrete question I have is: how does the long-term behavior of the learned dynamics look like, do we get chaotic dynamics if we let the system run freely for long periods of time or does it break down after a while and reach fixed point or limit cycle dynamics. the paper shows some long-term predictions, it doesn't address the stability of these predictions over very long time horizons
>
> We added a discussion of the long term stability of the Brusselator's system in the manuscript with : Figure 6 illustrates the performance of each trained model on the Brusselator system. After training on a specific time interval, we extended the integration period to five times the original training duration to evaluate the models' long-term behavior. We see that NDDE along with LSTM, Latent ODE are the only one's stable over this time interval. NDDE matches best over the long time horizon. Concerning the KS system, we do provide pdf estimations from the testset predictions.
>
> > if using only MSE, then at least better justification of that decision would be warranted
>
> We use the MSE to train our models, but we do assess its performance with additional evaluations. For example, the pdf plots for KS system and a new ablation study for the Cavity problem.

---

> > ### Author Response · Authors · 2024-11-13
> >
> > > Limitations and Future Work: more in-depth discussion of potential limitations or scenarios where the method might not perform well would balance it, potential limitations I could think of are :
> > > - computational complexity as the number of delays increases
> > > - challenges in determining the optimal number of delays
> > > - potential problems in high-dimensional delay spaces, or in very noisy settings
> > > - limitations in capturing very long-term dependencies
> > > - there's some discussion of computation, but more clear quantitative comparison would help
> > > - could add a few sentences on future directions
> >
> > In Appendix E.3, we present quantitative benchmarks on time and memory usage for a specific neural network size with varying numbers of delays. For a more theoretical examination of computational complexity, we add a reference to Appendix A.6 of this paper (https://arxiv.org/pdf/2306.14545) in Appendix E. The topic of determining optimal delays is briefly discussed in the second paragraph of our conclusion. Identifying the ideal number of delays is an unresolved issue, as reviewed here (https://arxiv.org/pdf/2302.03447) and we don't claim to solve any of this issue in the paper. However, we emphasize in our conclusion that overestimating delays does not adversely affect model performance. For instance, fitting the Brusselator system with an 8-delay NDDE yields similar results to using a 2-delay NDDE. In high-dimensional spaces, more delays also do not impair performance, as noted in the second paragraph of the conclusion. Regarding noisy settings, the last experiment involves a noisy dataset, demonstrating that NDDEs are robust to noise.
> >
> > > could add a few sentences on future directions
> >
> > Some future directions idea were given in the conclusion. We expand on other promising directions like comparing and studying the difference between the recently proposed Neural Fractional DE (https://arxiv.org/abs/2403.02737) and NDDEs.
> >
> > > Related Work: why not include reservoir computing for partially observed systems as related work? seems to be still close to state-of-the-art and is fast and easy to set up and train
> >
> > We wish to focus our paper on continuous time models since Neural DDE is one of those thus excluding other types of discrete models like Reservoir computing.  LSTM can be seen as an alternative to reservoir, so there is no real added value in having another comparison. Reservoir computing has a Markovian architecture, but accounts for correlations in time. In this sense, it is often compared to LSTM. Among the pros is true that it is rather fast to learn and for what concern the output gate even convex. However, it requires a rather large reservoir for relatively small systems. In this sense, LSTMs are "cheaper" in terms of architecture. Moreover, both are discrete-in-time.
> >  We can also refer to https://www.sciencedirect.com/science/article/abs/pii/S0893608020300708. In the abstract is stated "We find that, when the full state dynamics are available for training, Rerservoir Computing outperforms BPTT (i.e. backprop RNN/LSTM) approaches in terms of predictive performance and in capturing of the long-term statistics, while at the same time requiring much less training time."
> >
> > > why not include other NDDE approaches as comparison (Oprea, Schlaginhaufen)? What is precisely the difference to Oprea et al. 2023?
> >
> > Oprea et al's work and ours was developed concurrently but our approach is demonstrated on more challenging systems and with much larger networks (the network's used in Oprea has less than 10 parameters and all of the weights are initialised near their optimal values). Moreover, the adjoint is for one delay DDEs and their derivation of the adjoint is different compared to ours where we use Lagrangian multiplier. The main difference with Schlaginhaufen's is that we use a MZ formulation and possess learnable and multiple delays.
> >
> > > Additional Suggestions: more real-world applications as in climate science or biology would benefit the paper
> >
> > Our last experiment utilizes a dataset from a real world settings and the Brusselator system is linked to the field of chemistry/biology. Moreover, the cavity dataset is noisy which represents a real setting : unknown delays, noisy, real data. Very large actual systems is future work and considered now.
> >
> > > systematic study of performance under different noise levels would be valuable
> >
> > This would prove a valuable addition but we let this to future work.

---

> > > ### Author Response · Authors · 2024-11-13
> > >
> > > > there are geometric measures that could be helpful for analysing long-term dynamical behavior, e.g., Koppe et al. 2019 "Identifying nonlinear dynamical systems via generative recurrent neural networks with applications to fMRI"
> > >
> > > To the best of our understanding, Koppe et al 2019. uses variational inference (VI) to train its PLRNN-SSM model. Thanks to such training, some metrics like the KL divergence can be derived. Our model is simply trained via a L2 loss but doesn't utilize VI. Using VI with a Neural DDE would be a valuable addition to the paper but could be added as promising future directions. Indeed, very few papers use VI with Neural ODE like models, one example of this is https://proceedings.neurips.cc/paper_files/paper/2019/file/99a401435dcb65c4008d3ad22c8cdad0-Paper.pdf
> > >
> > > > but more generally, more metrics would paint a more complete picture - e.g., power spectrum correlation (especially for periodic systems), comparing lyapunov exponents (for chaotic) (in general an analysis of the prediction horizon would be interesting), or state space reconstruction metrics as the geometric measure mentioned above
> > >
> > > To further the discussion about stability we added the estimation of the maximum Lyapunov exponent for the KS system. We provide the maximum Lyapunov exponent (MLE) for our learned models of the KS system and provided the results in Section 4.2.
> > >
> > > > RNNs are critiqued as being discrete, but approximation used here is also discrete (or did I misunderstand?)
> > >
> > > Neural DDE with learnable delay parameters are by construction continuous-time models (one can evaluate the model at any given time t)
> > >
> > > > Presentation: graphical abstract or high-level diagram of the main concept (NDDEs for partially observed systems) would make it much easier to understand quickly what the paper is about
> > >
> > > We hope the update Figure 1 will help to understand better.
> > >
> > > > algorithm or pseudocode for NDDE may make the method more clear
> > >
> > > We inserted the NDDE algorithm 1.
> > >
> > > > table 2: add "model performance (MSE)" or similar instead of just model names
> > >
> > > Changed the title of table 2 to : "Model performance (MSE) over the test set on each experiments averaged over 5 runs"
> > >
> > > > figure 1: captions are quite small, also figure 4, 5 the labels should be larger
> > > > figure 3: add more context about what this represents
> > >
> > > This has been rectified
> > >
> > > > how sensitive is the NDDE performance to the choice of hyperparameters, particularly the number of delays? how to choose it? → sensitivity analysis for key hyperparameters would be helpful.
> > >
> > > This was mentioned in the conclusion. Adding more delays do not hurt overall performance of the models.
> > >
> > > > it's not clear to me how learning the delays works, how is this done in practice?
> > >
> > > Learning the delays are done via the adjoint method mentioned in Theorem 3.1. Yet we would note that if stability issues would arise with any type of system, they would first foremost arise in the data generation process before the training of NDDEs.
> > >
> > > > setup for partial observability for KS is not clear to me: how are the observed variables are chosen and in general how does this choice impact the results?
> > >
> > > The setup is described in the Section 4.1. We have chosen arbitrarily chosen 5 features that are evenly spaced on the $x$ axis of the KS system.
> > >
> > > > are there any numerical stability issues that can arise when solving delay differential equations, especially for chaotic systems like KS?
> > >
> > > Standard DDE stability issues arise as usual, for instance if time steps are taken too large, or equivalently accuracy thresholds are too wide, and that they are classically solved as well, that is, by considering smaller time steps or accuracy thresholds. If not followed some integration issues might arise.

---

> > > > ### Comment · Reviewer_JLGw · 2024-11-15
> > > > **Major improvement in clarity and presentation, still missing reservoir computing in related works and more context on Oprea.**
> > > >
> > > > Thank you for your thorough response. You have effectively addressed many of my major concerns, particularly regarding theoretical clarifications and experimental validations. The enhanced introduction updated theorems, and additional analyses significantly strengthen the paper.
> > > >
> > > > Your justification for excluding reservoir computing from benchmarks is reasonable, given your focus on continuous-time models. However, I still recommend including it in your related work section. The paper you cite (showing reservoir computing's superior performance with full-state dynamics) strengthens the need to discuss it in your literature review.
> > > >
> > > > While you've clearly articulated the differences between your approach and Oprea et al.'s work, I still recommend including a direct experimental comparison. Even if limited in scope, one could compare simpler systems where both approaches are applicable. This would help quantify the advantages of your larger networks and multiple delays. If a direct comparison is truly infeasible, then consider including a more detailed technical comparison in a table or dedicated subsection and provide specific examples of the "more challenging systems" your approach can handle.
> > > >
> > > > Overall, I think the paper has majorly improved in presentation and clarity.

---

> > > > > ### Author Response · Authors · 2024-11-20
> > > > >
> > > > > We thank the reviewers for the rebuttal answers and the additional insights provided. Please find below the provided answers to the latest comments.
> > > > >
> > > > > > Your justification for excluding reservoir computing from benchmarks is reasonable, given your focus on continuous-time models. However, I still recommend including it in your related work section. The paper you cite (showing reservoir computing's superior performance with full-state dynamics) strengthens the need to discuss it in your literature review.
> > > > >
> > > > > We thank the reviewer for his response, a part in RC will be integrated into the related works part of the paper. We will put this new RC part in the manuscript at the end of the RNN paragraph ie " However, the discrete nature of RNNs collides with the appealing continuous formulation of the problem".
> > > > >
> > > > > "Often compared to LSTMs, Reservoir Computing (RC), can be regarded as a noteworthy alternative to RNNs for its efficient training and strong performance in capturing long-term statistics when full state dynamics are accessible (https://www.sciencedirect.com/science/article/abs/pii/S0893608020300708). RCs use a fixed, randomly connected recurrent neural network (the reservoir) to capture the dynamics of input data while training only the output layer. Once mapped, only a simple readout layer is needed to extract the reservoir's state and train it to achieve the desired output (https://pubmed.ncbi.nlm.nih.gov/12433288/, https://www.science.org/doi/10.1126/science.1091277)"
> > > > >
> > > > >
> > > > > > While you've clearly articulated the differences between your approach and Oprea et al.'s work, I still recommend including a direct experimental comparison. Even if limited in scope, one could compare simpler systems where both approaches are applicable. This would help quantify the advantages of your larger networks and multiple delays. If a direct comparison is truly infeasible, then consider including a more detailed technical comparison in a table or dedicated subsection and provide specific examples of the "more challenging systems" your approach can handle.
> > > > >
> > > > > We believe that Oprea's implementation is rather limited and thus comparing our approach is complicated. One example of this is by taking into account the first example (logistic delay equation from https://arxiv.org/pdf/2304.01329). If we initialize with values "far away" from its optimal setup (for example $\theta_0 = \theta_1 = 0.1$, $\tau=2.5$ and $l_r=0.01$ for stable training), Oprea's implementation doesn't converge. If we reproduce this simple example of 3 parameter model with our approach, it converges. Oprea's demonstrates that learning delays is feasible on 3 parameters models, for one trajectory and where the loss landscape is convex (cf Fig.3). Our paper instead doesn't have any convex loss landscapes, has 10000x larger models and deals with batches of trajectories with all our examples (brusselator, ks, cavity) and seems to converge better.

---

### Review · Reviewer_s6Ca · 2024-10-30

**Summary Of Contributions:**

This paper considers neural DDE with constant lag delays. By embedding it under the Mori-Zwanzig (MZ) formalism, the authors show that in the case where the full system follows an ODE, the dynamics of its observables can be represented in the form of an DDE with finite number of delays. Neural network-based method for estimating the DDE is provided, whose performance is evaluated using several synthetic datasets.

**Audience:**

Yes

**Broader Impact Concerns:**

No.

**Claims And Evidence:**

Yes

**Requested Changes:**

1. Please expand on the estimation details in Section 3 (see also #1 in Weakness). In particular, it would be helpful to outline the important steps in the training pipeline (e.g., Algorithm 1 in Rubanova et al. Latent neural ODE, 2019), respectively for the two training schemas, and add necessary details for the major components.

2. See #2 in Weakness.

3. Please add necessary experiments that can more comprehensively demonstrate the performance of neural DDE with constant delay. In addition, please also include Zhu et al. 2019 (namely, neural DDE with a single delay) as a benchmark. See also #3 in Weakness.

4. See #4 in Weakness. Please include necessary details of the experiments in the main, so that the reader can at least know how the experiments are set up. Please also include experiments/results that can demonstrate whether the proposed method can recover system dynamics from data (e.g., see set up in Section 4.1 in Rubanova et al. Latent neural ODE)

5. See #5 in Weakness. Please consider include some experiments using real datasets.

6. Appendix J, it's unclear what it means by ``to train our models we progressively feed them longer trajectory chunks". The authors should consider adding some brief description on how the samples are constructed/fed into the training pipeline.

7. Section 4.2, could the authors explain why it is the case "every model incorporates a form of memory into its architecture, _with the exception of NODE_"?

**Strengths And Weaknesses:**

**Strength**

1. The connection between neural IDE, DDE and ODE is largely clear by utilizing the MZ formalism.
2. The authors show that a DDE formulation is suitable for modeling partially observed states of a system whose full dynamics follow an ODE.
3. This paper extends the framework developed in Zhu et al (2021) that considers a single time delay, and such an extension can potentially be useful in accommodating more complex dynamics.


**Weakness**

1. The presentation of this work could be significantly improved. In particular, in its current version, many important details have been omitted. For example, considering that this paper aims to provide a method for learning neural DDE with constant lag delays, details regarding how different pieces are handled/the training pipeline should be included in the main text, rather than just saying that an API/package is provided. The latter cannot replace proper description of the methodology.

2. This is related to presentation. If I am not mistaken, in the context of this paper, being able to model system with partially observed state is an application of the DDE, in the sense that by Proposition 2.2, there exists an exact representation in the form of a DDE, however only when the full system evolves according to an _ODE_. However, this "application" nature is never explicitly mentioned in the introduction, and hence in Section 2.1 where the notion of partially observed dynamical system is first brought up, it becomes confusing in that the scope of the modeling task is unclear.

3.  The experiments in Section 4 could be enhanced. Arguably, given the pitch of the paper, experiments should be divided into two parts:
    * Compare the proposed Neural DDE approach (with constant lag delays) against some other modeling tools such as (latent) neural ODE, neural DDE with a single time delay (Zhu et al. 2021 ICLR), LSTM/Transformers, when the system can be FULLY observed; this is to showcase the capacity of the proposed multiple-delay neural DDE as a modeling framework for dynamical systems.
    * Ditto the above when systems are partially observed, to showcase its "representation" capacity.  In addition, what if the full system does not follow an ODE (and therefore the exact representation no longer holds), in which case, would neural DDE be off by much?

    Currently, there is only one really-toy dataset to verify the approach "functions" (essentially Part I of the experiments is completely missing), and several cherry-picked dynamical systems for Part II. In other words,  the authors have not provided adequate experiments showcasing the capacity of the proposed framework.

4. Important experimental details are missing from the main. For example, how are the samples constructed/what is the context window? what is the sample size? In addition, how is training/validation/test split done? Without these details, it's even hard to access whether the test loss provides any indication on how the model performs in terms of forecasting (or equivalently, extrapolation) the system dynamics.

5. All experiments are based on synthetic data whose dynamics arguably are somewhat cherry-picked. The authors probably should consider applying the neural DDE with multiple delays to real datasets, and compare the model performance against benchmark methods.

---

> ### Author Response · Authors · 2024-11-13
>
> > The presentation of this work could be significantly improved. In particular, in its current version, many important details have been omitted. For example, considering that this paper aims to provide a method for learning neural DDE with constant lag delays, details regarding how different pieces are handled/the training pipeline should be included in the main text, rather than just saying that an API/package is provided. The latter cannot replace proper description of the methodology.
>
> We added Algorithm 1 in the paper that outlines the training procedure for a Neural DDE with learnable constant delays along with its generalization for several delays in Appendix H.
>
> > This is related to presentation. If I am not mistaken, in the context of this paper, being able to model system with partially observed state is an application of the DDE, in the sense that by Proposition 2.2, there exists an exact representation in the form of a DDE, however only when the full system evolves according to an ODE. However, this "application" nature is never explicitly mentioned in the introduction, and hence in Section 2.1 where the notion of partially observed dynamical system is first brought up, it becomes confusing in that the scope of the modeling task is unclear.
>
> We changed the theorem 2.1 to not include any reference to Appendix B and provided a discussion on how to deal with the noise term $F$ that refers to Appendix B. Proposition 2.2 was also updated to give the explicit conditions for the dynamical system and the observable $g$.
>
> > The experiments in Section 4 could be enhanced. Arguably, given the pitch of the paper, experiments should be divided into two parts:
> Compare the proposed Neural DDE approach (with constant lag delays) against some other modeling tools such as (latent) neural ODE, neural DDE with a single time delay (Zhu et al. 2021 ICLR), LSTM/Transformers, when the system can be FULLY observed; this is to showcase the capacity of the proposed multiple-delay neural DDE as a modeling framework for dynamical systems. Ditto the above when systems are partially observed, to showcase its "representation" capacity. In addition, what if the full system does not follow an ODE (and therefore the exact representation no longer holds), in which case, would neural DDE be off by much? Currently, there is only one really-toy dataset to verify the approach "functions" (essentially Part I of the experiments is completely missing), and several cherry-picked dynamical systems for Part II. In other words, the authors have not provided adequate experiments showcasing the capacity of the proposed framework.
>
> This paper specifically studies partially observed system which are much more challenging compared to fully observed systems.  Thus this makes fully observed systems out of the scope for this paper and not our focus (a fully observed system could be fitted with a Neural ODE). However, the present NDDE approach certainly applies to fully observed systems, while being over-expressive in that case. Regarding the datasets, the KS system is an established benchmark ((https://eudml.org/doc/51996, https://www.researchgate.net/publication/318868462_Hidden_Physics_Models_Machine_Learning_of_Nonlinear_Partial_Differential_Equations, https://arxiv.org/pdf/2308.05732, https://arxiv.org/pdf/2203.15706) for dynamical systems and the last experiment comes from real world experimental wind tunnel measurements.
>
> > Important experimental details are missing from the main. For example, how are the samples constructed/what is the context window? what is the sample size? In addition, how is training/validation/test split done? Without these details, it's even hard to access whether the test loss provides any indication on how the model performs in terms of forecasting (or equivalently, extrapolation) the system dynamics.
>
> This has been amended in the manuscript. $1024$ samples were generated for the Brusselator system, $2048$ for the KS system, $256$ for the toy dataset and the cavity is given in its citation source. The dataset splitting is train/val/test 70, 10, 20\%.
>
> > All experiments are based on synthetic data whose dynamics arguably are somewhat cherry-picked. The authors probably should consider applying the neural DDE with multiple delays to real datasets, and compare the model performance against benchmark methods.
>
> The KS system is an established benchmark (please refer to cited papers above) for dynamical systems and the last experiment comes from real world experimental wind tunnel measurements.

---

> ### Author Response · Authors · 2024-11-13
>
> **Requested changes**
>
> > Please expand on the estimation details in Section 3 (see also \#1 in Weakness). In particular, it would be helpful to outline the important steps in the training pipeline (e.g., Algorithm 1 in Rubanova et al. Latent neural ODE, 2019), respectively for the two training schemas, and add necessary details for the major components.
>
> To the request of the reviewer, we provide Algorithm 1 to showcase how a Neural DDE with one delay is trained in Section 3 and in the Appendix H for multiple delays.
>
> > This is related to presentation. If I am not mistaken, in the context of this paper, being able to model system with partially observed state is an application of the DDE, in the sense that by Proposition 2.2, there exists an exact representation in the form of a DDE, however only when the full system evolves according to an ODE. However, this "application" nature is never explicitly mentioned in the introduction, and hence in Section 2.1 where the notion of partially observed dynamical system is first brought up, it becomes confusing in that the scope of the modeling task is unclear.
>
> We changed the theorem 2.1 to not include any reference to Appendix B and provided a discussion on how to deal with the noise term that refers to Appendix B. Proposition 2.2 was also updated to give the explicit conditions for the dynamical system and the observable .
>
> > Please add necessary experiments that can more comprehensively demonstrate the performance of neural DDE with constant delay. In addition, please also include Zhu et al. 2019 (namely, neural DDE with a single delay) as a benchmark. See also \#3 in Weakness.
>
> It is logical to compare with the work of Zhu et al. since it is a prior study in the same field. It's important to mention that our approach includes the initial iteration of Neural DDEs. For instance the single fixed delays NDDEs in the cavity ablation experiment can be defined as Zhu's Neural DDE instantiation.
>
> > See \#4 in Weakness. Please include necessary details of the experiments in the main, so that the reader can at least know how the experiments are set up. Please also include experiments/results that can demonstrate whether the proposed method can recover system dynamics from data (e.g., see set up in Section 4.1 in Rubanova et al. Latent neural ODE)
>
> This is amended in the manuscript.
>
> > See \#5 in Weakness. Please consider include some experiments using real datasets.
>
> Regarding the datasets, the KS system is an established benchmark (please refer to cited papers above) for dynamical systems and the last experiment comes from real world experimental wind tunnel measurements.
>
> > Appendix J, it's unclear what it means by ``to train our models we progressively feed them longer trajectory chunks". The authors should consider adding some brief description on how the samples are constructed/fed into the training pipeline.
>
> We added extra information to complete Appendix J and we hope that Algorithm 1 elucidates the training procedure.
>
> > Section 4.2, could the authors explain why it is the case "every model incorporates a form of memory into its architecture, with the exception of NODE"?
>
> Neural ODE is purely Markovian, it has no memory, in contrast with ANODE, Latent ODE, LSTM, etc.

---

### Review · Reviewer_7qkz · 2024-10-30

**Summary Of Contributions:**

The authors explore to utility of neural delay differential equations with learnable delay sizes. They connect the approach of neural delay differential equations to the Mori-Zwanzig formalism describing the task providing a robust motivation for the family of methods. They then compare the DDE approach against several NODE approaches and RNNs with and without memory and find improved performance both from the general concept of the DDE and the specific implementation with learned delays.

**Audience:**

Yes

**Broader Impact Concerns:**

No concerns.

**Claims And Evidence:**

No

**Requested Changes:**

__Critical__
1. Update the paper to include clearer notation and argumentation. Remove unnecessary information and think about how the primary message of the paper can be streamlined. Ensure terms are defined prior to usage and that the reader understands the essentials for your arguments from the main text along. Technical details can be left to the appendix, but the core message should be understandable without opening the appendix.
2. Move the conditions for the formalism to hold to the main text. This is some of the most important information
3. Clean up figures. Make sure axes are informative and that the user knows what they're looking at. Fonts should be readable without zooming.
4. Run experiments on the same scale. For neural ODEs, this is probably run-time. If using adaptive time-steppers, it probably should include training time as well.
5. Note the figure pulled from another paper in the caption itself.

__Strengthen__

1. Explore some of the other questions this opens up. How do things like the history function impact the solution? How does varying the initial distribution of delays matter.
2. Use consistent backgrounds on plots. It's a bit strange to have both gray and white backgrounds.

**Strengths And Weaknesses:**

__Strengths__
1. The problem is well motivated and the proposed solution makes sense.
2. The authors effectively demonstrate the advantage of learned delays over constant delays under the initialization scheme.
3. The results are promising and seem to indicate that the proposed method outperforms several competitors.

__Weaknesses__

1. The major weakness right now is clarity and presentation. The presentation is often very imprecise. It jumps arount frequently, sometimes using notation that hasn't been defined yet, and links do not always succintly answer the information they promise. Plots have very light labeling. Sometimes it feels as though irrelevant information has been introduced. Examples:
    1. Equation 1 is using notation that has yet to be defined. We have L sets of N tuples. Is $N$ samples in time and $L$ samples in trajectories? If so, say this when you use them. Is $N$ connected to the $n$ used for dimension of the system? There are a limited number of symbols defined at this point, so reusing symbols would seem to have meaning, but it doesn't appear there is.
    2. What does it mean for the approach to be "valid"? Is it poorly constrained resulting in non-unique solutions? Describe why something is considered "invalid" specifically.
    3. Discussion about expressivity - again, this is quite vague. Is there a function class that one family of ODEs can solve but not another? How does memory address this issue? Even if it seems obvious from working in the space, it's important to explain the necessary details for the general audience to understand.
    4. Appendix B does not list the assumptions. These assumptions are absolutely vital to understanding when this approach applies. If they are in an appendix somewhere, they should be in the main text.
    5. Much of the information presented seems irrelevant. Specifics:
        1. The neural ODE, ANODE, and neural IDE equations are all written out explicitly in their own lines and again in a separate figure. Why? It the equation for ANODE important or is it just important to know that it's augmented with memory?
        2. The t-model is introduced, there is an appendix deriving it, but then it is never used. The methods in the paper do not seem directly related to the t-model. It seems like this could be mentioned very briefly in a related work section.
    6. In equation 5, it seems that the way $g_i$ have been defined would make the function $g(x(t))$. Later a specific $g(t)$ is described through it's relationship to $h$ which is a one-off definition and never used again.
    7. $G$ and $g$ again reuse the same letter, but seem to be disconnected. $g$ seems to be an observable mapping while $G$ is an arbitrary function. $g$ seems to be an observation of $x$ rather than $G$ so reusing the same letter as a symbol doesn't completely make sense here.
    8. Generally figures need more labeling and resizing - the text is too small to be readable in several. This is probably an optical illusion, but right now it looks like the lines in figure 1 are diagonal? Is horizontal then time and vertical is space?
        1. Figure 1 could also be more informative. It's somewhat clear that we're moving from the full state evolution to modeling a set of points, but that was already clear from the text. How is this specifically showing the MZ formalism?
        2. Figure 2 does not provide much information right now. It's a set of equations with a sliding scale. This could potentially replace text, but it's spelled out using an equivalent amount of space in text already so this is just redundant.
        3. The results need labels. Which fields are we looking at in the samples?
2. There are a larger number of questions opened, but not really explored. The proposed method introduces a number of hyperparameters. Number of delays is considered, but it's not clear how the initial distribution of these delays matters.
3. From appendix J, it's not clear whether these comparisons are fair. Parameter count varies considerably. Additionally, with neural ODE type approaches, parameter count is a very poor proxy for performance as the integrators and run-time can be heavily impacted by learning dynamics. It's not clear which integrators are used which also has a large impact on performance. Additionally, it seems that there are a few similar DDE variants in existence one of which is mentioned and the equation is provided. It would be more illustrative to see how the authors' proposed approach for learning delays compares to other approaches designed for delay differential equations. How were HPs tuned? A single LR is posted - was this tuned for the DDE and applied to others or was each model tuned separately?
4. Appendix J also introduces some new terminology that hasn't been seen before. What is "patience"?. In general, I would say the details in J are insufficient for someone to attempt reproducing the work. It would be beneficial to have more detailed explanation of what model architectures are being used both in the appendix and the main text - someone reading the main text should at least have a rough understanding of what models are being used within the neural ODEs.

---

> ### Author Response · Authors · 2024-11-13
>
> > The major weakness right now is clarity and presentation. The presentation is often very imprecise. It jumps arount frequently, sometimes using notation that hasn't been defined yet, and links do not always succintly answer the information they promise. Plots have very light labeling. Sometimes it feels as though irrelevant information has been introduced. Examples:Equation 1 is using notation that has yet to be defined. We have L sets of N tuples. Is samples in time and samples in trajectories? If so, say this when you use them. Is connected to the used for dimension of the system? There are a limited number of symbols defined at this point, so reusing symbols would seem to have meaning, but it doesn't appear there is.
>
> We would amend the introduction to be more explicit by stating : "In a data-driven context, given a dataset of $L$ distinct trajectories of length $N+1$, represented as $\{(t_0^l, x_0^l), \dots, (t_{N}^l, x_{N}^l) \}_{l=1}^{L}$, which are observations of a unknown system:"
>
> $N+1$ represents the number of time steps in each trajectory and $n$ represents the dimension of the state $x(t)$.
>
>
>
> > What does it mean for the approach to be "valid"? Is it poorly constrained resulting in non-unique solutions? Describe why something is considered "invalid" specifically.
>
> The introduction was changed from "Such an approach is valid if the system's state is fully observed and the system is Markovian. Neural Ordinary Differential Equations (NODEs), introduced in [Chen et al](https://arxiv.org/abs/1806.07366), follows this exact formulation. NODEs gave rise to the class of continuous depth models and can be viewed as a continuous extension of Residual Networks  [He et al](https://arxiv.org/abs/1512.03385)." to "Using Equation 1's ODE dynamics model to fit the aforementioned dataset is natural and well suited."
>
> > Discussion about expressivity - again, this is quite vague. Is there a function class that one family of ODEs can solve but not another? How does memory address this issue? Even if it seems obvious from working in the space, it's important to explain the necessary details for the general audience to understand.
>
> The notion of expressivity is the main focus of the Augmented Neural ODE (ANODE) paper (https://arxiv.org/pdf/1904.01681). ANODE discusses this in their section 3 and 4. In order to make this more explicitly, we propose to go from :
>
> "An immediate extension of NODEs, known as Augmented NODEs, addresses the expressivity limitation of NODEs by augmenting the dimension of the solution space from $\mathbb{R}^n$ to $\mathbb{R}^{n+p}$  through the introduction of an extra proxy variable $a(t)$. The additional introduced dimensions allow to learn more complex functions using simpler flows." to "An immediate extension of NODEs, known as Augmented NODEs discussed about the existence of some functions that NODEs cannot represent, thus addresses the expressivity limitation of NODEs by augmenting the dimension of the solution space from $\mathbb{R}^n$ to $\mathbb{R}^{n+p}$  through the introduction of an extra proxy variable $a(t)$. The additional introduced dimensions allow to learn more complex functions using simpler flows."
>
> >Discussion about expressivity - again, this is quite vague. Is there a function class that one family of ODEs can solve but not another? How does memory address this issue? Even if it seems obvious from working in the space, it's important to explain the necessary details for the general audience to understand.
>
> Considering the modifications made to the introduction, we hope that the discussion on expressivity first mentioned in the introduction of ANODE is now clearer.
>
> > Appendix B does not list the assumptions. These assumptions are absolutely vital to understanding when this approach applies. If they are in an appendix somewhere, they should be in the main text.
>
> We changed the theorem 2.1 to not include any reference to Appendix B and provided a discussion on how to deal with the noise term $F$ that refers to Appendix B. Proposition 2.2 was also updated to give the explicit conditions for the dynamical system and the observable $g$.
>
> > The neural ODE, ANODE, and neural IDE equations are all written out explicitly in their own lines and again in a separate figure. Why? It the equation for ANODE important or is it just important to know that it's augmented with memory?
>
> Figure 2 in the first version of the paper is used to provide an overview/summary of the modeling choices possible for the MZ formalism and to appreciate the different functional structures of the different models. This remark has been mentioned by another reviewer and we decided to remove it.

---

> > ### Author Response · Authors · 2024-11-13
> >
> > > The t-model is introduced, there is an appendix deriving it, but then it is never used. The methods in the paper do not seem directly related to the t-model. It seems like this could be mentioned very briefly in a related work section.
> >
> > Introducing the t-model was part of enumerating the different known approximations in the literature. The t-model itself can be modeled via a time-dependent ODE. Amendments were made to incorporate this and to make the presentation clearer. "For example, the t-model, also commonly called slowly decaying memory approximation (Chorin et al, 2002) leads to Markovian equations with time-dependent coefficients (see Appendix C for full derivation)." to "For example, the t-model, also commonly called slowly decaying memory approximation (Chorin et al, 2002) leads to Markovian equations with time-dependent coefficients , which can subsequently be modeled using a NODE (see Appendix C for full derivation).
> >
> > > In equation 5, it seems that the way $g_i$ have been defined would make the function $g(x(t))$. Later a specific $g(t)$ is described through it's relationship $h$ to which is a one-off definition and never used again.
> >
> > We thank the reader for this further simplification suggestion, $h$ was removed from Figure 1.
> >
> > > $G$ and $g$ again reuse the same letter, but seem to be disconnected. $g$ seems to be an observable mapping while $G$ is an arbitrary function. $g$ seems to be an observation of $x$ rather $G$ than so reusing the same letter as a symbol doesn't completely make sense here.
> >
> > In order to avoid confusion in Theorem 2.1, $G, g$ was changed to $L,l$ which makes more sense here because it defines a loss function.
> >
> > > Generally figures need more labeling and resizing - the text is too small to be readable in several. This is probably an optical illusion, but right now it looks like the lines in figure 1 are diagonal? Is horizontal then time and vertical is space?
> >
> > We modified Figure 1 to add the x and y axis to provide more information. We also added more description to how the figure is showing the MZ formalism. The caption was changed from "The MZ equation DDE approximation used to model partially observed systems." to "Fully observed systems (1) due to their high dimensionality and sparse observations can only be seen through the lens of an observable function $g$ (2). Ultimately the user only has at its disposable the dynamics of $g(t)$. The MZ equation DDE approximation (Proposition 2.2) is then used to model partially observed systems (3)."
> >
> > > Figure 2 does not provide much information right now. It's a set of equations with a sliding scale. This could potentially replace text, but it's spelled out using an equivalent amount of space in text already so this is just redundant.
> >
> > We removed Figure 2.
> >
> > > The results need labels. Which fields are we looking at in the samples?
> >
> > In the first paragraph of Section 4.2, we do specify the nature of the labels. To add more explicit content we decide to change the last sentence from Section 4.2 "In all subsequent figures, the y-axis $y(t)$ represents our observables, defined as $y(t) = g(x(t))$."  to  "In all subsequent figures, the y-axis $y(t)$ represents our observables (introduced in Section 4.1) , defined as $y(t) = g(x(t))$."
> >
> > > There are a larger number of questions opened, but not really explored. The proposed method introduces a number of hyperparameters. Number of delays is considered, but it's not clear how the initial distribution of these delays matters.
> >
> > Figure 10 provided an ablation study for fixed and learnt delays. We decided to improve this experiment by experimenting with different delay initialization values and added our insights in the revised paper in Section 4.2 in the cavity paragraph.
> >
> > > From appendix J, it's not clear whether these comparisons are fair. Parameter count varies considerably. Additionally, with neural ODE type approaches, parameter count is a very poor proxy for performance as the integrators and run-time can be heavily impacted by learning dynamics. It's not clear which integrators are used which also has a large impact on performance. Additionally, it seems that there are a few similar DDE variants in existence one of which is mentioned and the equation is provided. It would be more illustrative to see how the authors' proposed approach for learning delays compares to other approaches designed for delay differential equations. How were HPs tuned? A single LR is posted - was this tuned for the DDE and applied to others or was each model tuned separately?
> >
> > We tried our best to have a fair comparison with all of the models. Concerning all the initial value problem models (NODE, ANODE, NDDE, Latent ODE), we decided to give it the same model architecture. We also used a high order solver, RK4. For LSTM, we matched its parameter count. Regarding the LR, no extensive tuning has taken place and thus other learning rates might be more optimal than the one used for the other models and NDDEs.

---

> > > ### Author Response · Authors · 2024-11-13
> > >
> > > > Appendix J also introduces some new terminology that hasn't been seen before. What is "patience"?. In general, I would say the details in J are insufficient for someone to attempt reproducing the work. It would be beneficial to have more detailed explanation of what model architectures are being used both in the appendix and the main text - someone reading the main text should at least have a rough understanding of what models are being used within the neural ODEs.
> > >
> > > We provide more information in the appendix of the "patience". We put in Appendix J "Our model training approach incorporates a progressive strategy inspired by curriculum learning (https://arxiv.org/abs/2101.10382). We begin by feeding the models shorter trajectory segments and gradually increase their length when the patience hyperparameter is exceeded. This process continues until we reach the desired trajectory length. Each time the trajectory length is increased, we reset the patience hyperparameter to 0. It is then incremented by 1 if the validation loss fails to decrease, and reset to 0 if the validation loss improves. This method aligns with the principles of curriculum learning, a technique that involves training machine learning models in a structured order, typically progressing from simpler to more complex examples. In our case, this translates to moving from shorter to longer trajectories. This approach aims to enhance the learning process and improve model performance.
> > >
> > > **Requested Changes**
> > >
> > > > Update the paper to include clearer notation and argumentation. Remove unnecessary information and think about how the primary message of the paper can be streamlined. Ensure terms are defined prior to usage and that the reader understands the essentials for your arguments from the main text along. Technical details can be left to the appendix, but the core message should be understandable without opening the appendix.
> > >
> > > This was amended in the manuscript with the previous comments.
> > >
> > > > Move the conditions for the formalism to hold to the main text. This is some of the most important information
> > >
> > > This was amended in the manuscript with the previous comments.
> > >
> > > > Clean up figures. Make sure axes are informative and that the user knows what they're looking at. Fonts should be readable without zooming.
> > >
> > > This was amended in the manuscript with the previous comments.
> > >
> > > > Run experiments on the same scale. For neural ODEs, this is probably run-time. If using adaptive time-steppers, it probably should include training time as well.
> > >
> > > In the discussion regarding Appendix J, we do believe that each approaches are comparable.
> > >
> > > > Note the figure pulled from another paper in the caption itself.
> > >
> > > This was fixed
> > >
> > > > Explore some of the other questions this opens up. How do things like the history function impact the solution? How does varying the initial distribution of delays matter.
> > >
> > > This was amended in the manuscript with the previous comments. For dynamical system fitting, the history function will always be a small portion of the trajectory (cf line 2 of Algorithm 1) in order to correctly initialize Neural DDEs.
> > >
> > > > Use consistent backgrounds on plots. It's a bit strange to have both gray and white backgrounds.
> > >
> > > This was fixed

---

### Author Response · Authors · 2024-11-13

We thank the reviewers for their insights. By taking into account your comments, we made some changes to the article (that is now updated on openreview). The main new changes are :
- Modification of Figure 1 that explains a bit more in details the MZ framework with DDEs.
- Removal of Figure 2 since it seemed to be considered redundants.
- Added Algorithm 1 that shows how to train Neural DDE with learnable delays
- Modified Theorem 2.1 to not make a reference to Appendix B and updated Proposition 2.2
- Long term stability analysis with Maximum Lyapunov exponent estimation (Table 3) for the KS system and by visualizing a long rollout of the Brusselator system (Figure 6)
- Enhanced the ablation study of fixed and learnt delays for the cavity dataset by also looking at the influence of the delay initialization (Figure 10).

Other changes are also amended in the manuscript are discussed in the personal responses for reviewers.

---

### Decision · Action_Editor_rnfH · 2024-12-16

**Recommendation:** Reject

**Comment:**

I find the paper very interesting, however the reviewers are right in that, the experimental evaluation is not solid enough, there is no comparison with fixed delay DDE method. Also as the reviewers mentioned, a rework of the presentation could help the understandably significantly.
I would like to encourage the authors to resubmit after revision.

**Audience:**

Yes, I expect a considerable part of the audience would be interested.

**Claims And Evidence:**

Some of the empirical evidence is not solid enough, more relevant benchmarks (like fixed delay DDE methods need to be added).

**Resubmission Of Major Revision:**

The authors may consider submitting a major revision at a later time.